# Synergistic morphology and feedback control for traversal of unknown compliant obstacles with aerial robots

Emanuele Aucone [1,2] ✉, Christian Geckeler [1,2], Daniele Morra[3], Lucia Pallottino[3] & Stefano Mintchev [1,2]

Animals traverse vegetation by direct physical interaction using their entire body to push aside and slide along compliant obstacles. Current drones lack this interaction versatility that stems from synergies between body morphology and feedback control modulated by sensing. Taking inspiration from nature, we show that a task-oriented design allows a drone with a minimalistic controller to traverse obstacles with unknown elastic responses. A discoid sensorized shell allows to establish and sense contacts anywhere along the shell and facilitates sliding along obstacles. This simplifies the formalization of the control strategy, which does not require a model of the interaction with the environment, nor high-level switching conditions for alternating between pushing and sliding. We utilize an optimization-based controller that ensures safety constraints on the robot's state and dampens the oscillations of the environment during interaction, even if the elastic response is unknown and variable. Experimental evaluation, using a hinged surface with three different stiffness values ranging from 18 to 155.5 N mm rad$^{-1}$, validates the proposed embodied aerial physical interaction strategy. By also showcasing the traversal of isolated branches, this work makes an initial contribution toward enabling drone flight across cluttered vegetation, with potential applications in environmental monitoring, precision agriculture, and search and rescue.

Aerial robots perceive vegetation as obstacles to avoid[1,2]. However, dense yet compliant branches, twigs, and leaves could potentially be traversed using direct physical interaction. This would give access to currently unreachable areas, enabling the collection of valuable data for environmental monitoring[3], precision agriculture[4], and search and rescue[5,6].

Despite considerable advancements over the last decade[7], the design and control of aerial robots still face limitations when interacting with compliant obstacles. Common approaches for aerial physical interaction (APhI), such as impedance and admittance controllers, are easy to implement but are mainly tailored for exerting desired forces on rigid surfaces[8–10]. In contrast, traversing vegetation necessitates the use of different interaction modes, including pushing to bend obstacles, sliding along them, or employing a combination of these techniques. Model-based or robust controllers enable different modes of interactions as demonstrated in complex tasks such as making contact and pushing hinged doors or rolling carts[11–13], and pushing and sliding along surfaces for writing or inspection[14–16]. However, these solutions require high-level switching policies based on empirically tuned conditions and thresholds, which become complicated to define when mechanical properties of the environment are stochastic and complex to model, as is the case with vegetation. Moreover, while

[1]Environmental Robotics Laboratory, Department of Environmental Systems Science, ETH Zürich, Zürich, Switzerland. [2]Swiss Federal Institute for Forest, Snow and Landscape Research, WSL, Birmensdorf, Switzerland. [3]Research Center "E. Piaggio", Department of Information Engineering, University of Pisa, Pisa, Italy. ✉e-mail: eaucone@ethz.ch

extensive research has been dedicated to APhI with rigid and movable obstacles[17], obstacles with an elastic response, such as vegetation, have received limited attention. Another limitation of current drones is that interaction tasks are constrained to sensorized end-effectors, as demonstrated by tasks such as sensor installation and retrieval[18,19], contact-based inspection[20,21], or object manipulation[22,23]. However, traversing vegetation demands an "unconstrained" interaction, where contacts can be established and moved along the entire body of the drone during traversal. As a result, haptic sensing cannot be confined to the end-effector but must be distributed along the entire body of the robot.

Animals have evolved remarkable biomechanical and locomotion strategies[24–27], allowing them to move safely in cluttered vegetation. Through a synergistic interaction of body morphology and feedback control modulated by sensing, animals can transition between various locomotor and interaction modes for navigating complex terrains[28,29]. For example, cockroaches traverse flexible, grass-like beams by initially pushing across the beams and often rolling their bodies to efficiently slide through the gaps[27]. This transition is the result of the interplay between their terradynamically streamlined body[26] and sensory feedback control. Some insects have also evolved microstructured body surfaces to reduce the friction when moving under leaves or burrowing[30,31]. Translating these insights in task-oriented morphology and minimalistic controllers, researchers are developing ground robots that use their sensorized body to skillfully push aside and slide through compliant obstacles[26,27,32], drones that exploit protective cages to fly through rigid obstacles[33,34], and foldable drones that traverse narrow passageways[35,36].

The locomotor and interaction versatility demonstrated by animals can inspire the development of novel embodied APhI strategies that tightly integrate morphology, sensing, and feedback control. In this study, we present a direct physical interaction strategy designed for the traversal of a single obstacle with a large and unknown range of stiffnesses, utilizing an underactuated aerial robot (Fig. 1 and Supplementary Movie S1). Our embodied APhI strategy integrates a task-oriented morphology with a simplified yet versatile feedback controller modulated by distributed haptic sensing. A sensorized discoid shell enables the making and sensing of contacts at any point along the body, and facilitates sliding over obstacles due to the streamlined shape and low-friction surface. These design features simplify the control strategy by eliminating the need for a contact model, constraints, or complex switching conditions between pushing and sliding, thereby reducing the computational load. Indeed, we utilize a straightforward optimization-based controller that uses force feedback to maintain robot safety by dampening environmental oscillations during interactions, without requiring knowledge about the

elastic response. In a series of experiments, we demonstrate the feasibility of our approach as the drone successfully traverses hinged compliant plates, validating its effectiveness for three different stiffness values spanning one order of magnitude. Through an ablation study, we show that the task-oriented morphology and sensory driven control feedback lead to successful traversal of the obstacle only when exploited simultaneously. Furthermore, additional experiments using real branches with and without leaves confirm the versatility of our approach and provide insights into the challenges of real-world applications.

## Results

### Task definition

We study the task of traversing a single compliant obstacle consisting of a rigid plate connected to a vertical hinge with a torsional spring of stiffness K. Different obstacle stiffness levels are achieved by varying the value K, which is unknown to the robot. Given the geometry of both the obstacle and the drone, we assume that the interaction problem predominantly occurs on a plane (Fig. 2).

Figure 2 illustrates the task and the different interaction modes that the drone needs to alternate in order to traverse the obstacle. Starting from a hovering condition, the drone follows a straight reference path toward the obstacle, then deflecting it away to fly past it (Fig. 2A). Upon contact with the obstacle, the drone transitions from a zero to a non-zero interaction wrench (consisting of the forces and torques exchanged between the drone and the environment during the interaction); strong impacts may destabilize the robot. To deflect the obstacle, the drone has to start Pushing. In this mode (Fig. 2B), the relative movement between the drone and the obstacle is minimal, as the drone maintains a static point of contact while the obstacle is pushed. The Sliding mode instead, occurs when the obstacle cannot be moved (Fig. 2C), e.g., because it is too stiff, causing the robot to slide on it. The drone can actively slide along the surface of the obstacle without losing contact. In this mode, relative movement occurs between the moving drone and the static obstacle. The two interaction modes may coexist in the Push-and-Slide mode (Fig. 2D), that occurs when the drone pushes the obstacle away while simultaneously sliding on it. Here, the point of contact moves on both the obstacle's surface and the robot's body. During the whole interaction, the drone maintains contact with the surface, i.e., no loss of contact. Upon detachment, the drone loses contact with the obstacle and may be subject to an abrupt change in the interaction wrench.

### Design rationale

Robots traversing a compliant obstacle by physical interaction can derive benefit from environment- and task-oriented morphologies,

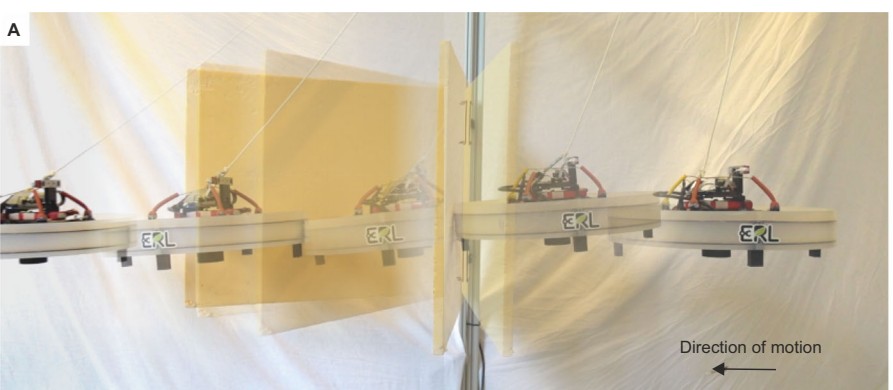

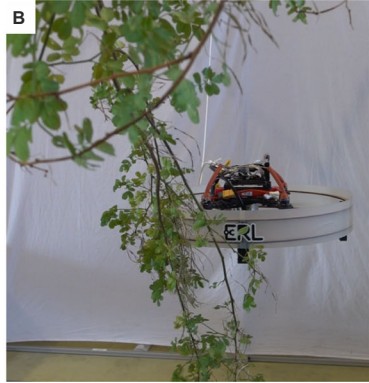

Direction of motion

**Fig. 1 | Traversal of complaint obstacles with a drone.** An underactuated aerial robot equipped with a discoid shell traversing a compliant, hinged plate whose dynamic parameters are unknown. The robot is required to follow a desired path during the interaction, which translates into the need to actively push and slide to overcome the obstacle. Experimental validation conducted with (**A**) a hinged plate and (**B**) a real branch.

facilitating task execution and simplifying the formulation of feedback control systems. We have equipped a quadrotor with a disc-shaped shell that protects the propellers from collisions and can be used to push and slide on the compliant plate (Fig. 3A, manufacturing details available in the "Methods" section). The circular shape provides symmetry to the robot, prevents any discontinuities in the contact point while interacting, and generalizes the interaction in any direction on the plane defined by the disc. The low-friction fiberglass surface of the shell further promotes the sliding of the obstacle along its surface. To enable force feedback control, we integrated a six-axis load cell connecting the quadrotor's frame and the shell. The result is a distributed haptic sensor that measures the net wrench resulting from contacts occurring across the entire surface the shell.

## Controller rationale

A sensory feedback controller allows the robot to follow a reference path while safely interacting with compliant obstacles of unknown stiffness. We utilize a Nonlinear Model Predictive Control (NMPC) framework formalized in the drone's center of mass (CoM) (1) to take into account the full dynamics of the drone and how they are affected by the external wrench exerted by the obstacle, (2) to introduce specific cost terms in order to achieve the different objectives of following a path and simultaneously interacting with the compliant obstacle, and (3) to impose constraints on the drone's dynamics to guaranteeing safe operation. In the following sections, we examine these three features of the controller. The mathematical formulation and implementation details are comprehensively described in the "Methods" chapter and in Supplementary Method 3. Figure 3B portrays the overall control scheme, which involves the proposed NMPC-based strategy embodied in a standard feedback loop of an aerial robot.

The NMPC takes into account the full model of the dynamics of the drone, including the external wrench in both the translational and rotational dynamics. This is favorable to predict how the measured external wrench will affect the robot's dynamics.

Two objective terms are introduced in the NMPC to enable the movement toward the other side of the obstacle and to simultaneously interact with it. The first objective term is for path following and commands the drone to follow a straight path and to stay as close as possible to it, tracking a desired velocity. The second objective term is for physical interaction and consists of an impedance behavior included in the NMPC optimization to shape the desired response of the drone while in contact with the obstacle. A similar approach has been validated for both manipulators[37] and aerial robots[38], but never for interaction with compliant environments. By defining a desired impedance, the controller dampens oscillations of the elastic environment without the need for a model. This approach is independent of the location of the point of contact, allowing us to exploit the benefits of the design, i.e., interaction on every point on the cage and unconstrained with respect to the point of contact. Properly balancing the two objectives allows achieving good performance, as demonstrated in our ablation study on the impact of the controller parameters on the behavior of the drone during the interaction (see Supplementary Method 6 and Supplementary Movie S5).

Finally, a set of safety constraints in the NMPC creates guarantees to avoid oscillations upon contact and detachment, as well as to permit a smooth and direct transition between the interaction modes thanks

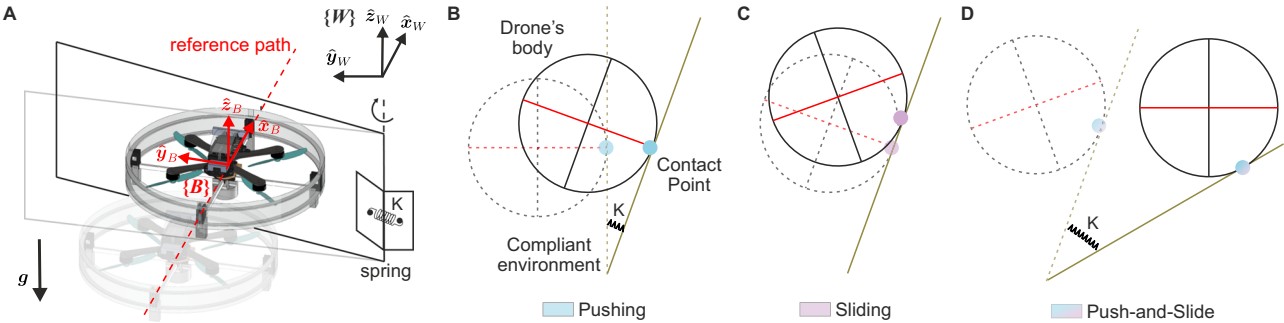

**Fig. 2 | Task and modes of interaction. A** Visual representation of the robot following a reference path while interacting with a compliant environment (a hinged surface with compliance around the anchor point). Highlighted: gravity; world, inertial $W$ frame; and body $B$ frame centered and fixed in the drone's center of mass. Top-view schematic of (**B**) Pushing, (**C**) Sliding and (**D**) Push-and-Slide phases. K is the torsional stiffness of the compliant obstacle.

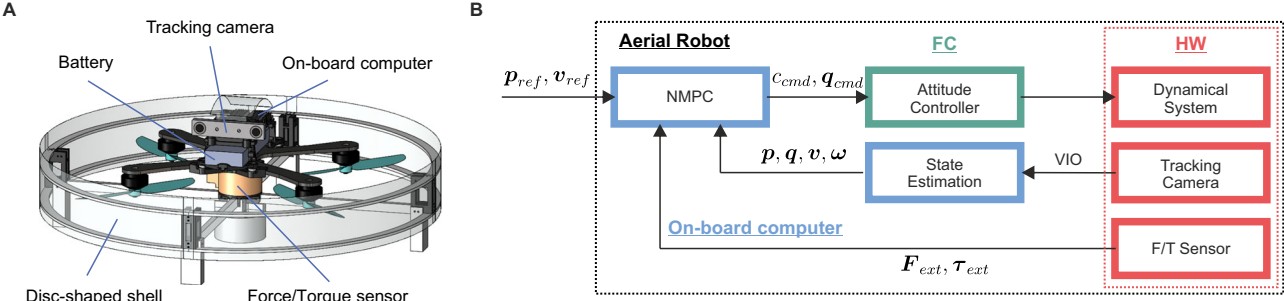

**Fig. 3 | Task-oriented design of the aerial robot. A** CAD of the aerial robot, consisting of a quadrotor connected to a streamlined, disc-shaped shell via a force/ torque (F/T) sensor. **B** Block diagram of the control framework: the full odometry (position $p$, orientation $q$, linear $v$ and angular $\omega$ velocities) is provided by the on-board tracking camera (VIO) and converted into the correct frame by the state estimator, while the external wrench ($F_{ext}$, $\tau_{ext}$) is provided by the force/torque sensor that connects the quadrotor and the cage. Starting from desired position and velocity ($p_{ref}$, $v_{ref}$), which define the path to follow, the controller (NMPC) outputs a command of mass-normalized thrust $c_{cmd}$ and orientation $q_{cmd}$ to the low-level, Attitude controller running on the Flight Controller (FC).

to the body shape. In detail, enforcing constraints on the dynamics of the drone (e.g., max pitch and roll angles), in combination with flying at a low reference speed, allows us to assure a quasi-static motion and a safer operation, by limiting the external wrench exchanged during interaction. Increasing speed would result in higher wrenches exerted on the drone, which would make the traversal more challenging (e.g., difficulty in smoothly transitioning between interaction modes due to the high speed and short interaction time) or destabilize the drone (e.g., drone not capable of counteracting a high wrench due to physical limitations of the platform).

The task-oriented design enables substantial simplifications in the feedback controller. APhI generally requires an analytical description of the environment and, most importantly, a set of models and constraints related to the contact point, in order to define the motion of the end-effector and the interaction with the environment (e.g., end-effector partially or fully constrained in position and/or orientation, assumption of rigid contact with the environment, friction cone contact models). Such constraints are necessary to define the switching logic for transitioning between interaction modes. Thanks to the streamlined shape of the shell and the use of low-friction materials, the proposed controller does not need constraints on the contact point, which can occur and move anywhere on the shell. This is beneficial because (1) the implementation of the controller is simplified, (2) the controller has less constraints so it can find a feasible solution faster, and (3) the controller can exploit the robot's body to automatically transition between Pushing, Sliding, and Push-and-Slide, which will be induced by the motion itself, without imposing high-level strategies to handle them. Our controller only requires measuring the response of the environment in order to predict how the robot dynamics will be affected, without requiring the location of the contact point or a model of the elastic environment. In our scenario, the stiffness of the obstacle is unknown, does not need to be estimated, and the controller can therefore generalize for different values of compliance. This is favorable because in natural environments obstacles can have highly

nonlinear dynamics, often unfeasible to model or estimate online. Furthermore, without the need for accurate contact dynamics, the controller can be released from any type of contact constraint, which generally increase the optimization problem's complexity.

## Experimental validation

We report the experimental validation of our traversal strategy for different values of stiffness (thus, compliance) of the obstacle, by changing the torsional spring at the hinge. In detail, we selected three values (18, 77.8, 155.5) N mm rad$^{-1}$, which we refer to as low, mid and high.

First, we report an experiment conducted with the mid value of stiffness, with the aim of analyzing the interaction behavior of the drone during the traversal. In Fig. 4A, the different phases of the traversal task are depicted with still frames from lateral and top views. The direction of motion is from right to left. In Fig. 4B, the plots show the time evolution of the variables of interest (attitude and external wrench) during the execution of the task. Upon first contact, the drone starts pushing against the hinged plate (light blue zone) and the force along the longitudinal axis $F_{\mathrm{ext},x}$ increases. Simultaneously, the drone undergoes rotation around the yaw axis ($\psi$) following the deflection of the obstacle. To follow the reference, straight path the drone begins to rotate to adjust the yaw angle in the opposite direction. The rotation is facilitated by the low-friction shell and the cost terms related to path following included in the objective function of the NMPC. Eased by the circular morphology of the shell and its low-friction surface, this action results in a Sliding interaction (light pink zone), which can be seen by the change in the force along the lateral axis $F_{\mathrm{ext},y}$, in the yaw angle $\psi$, and in the torque around the vertical axis $\tau_{\mathrm{ext},z}$. At this point, the drone starts to push again, in order to go straight, further deflecting the obstacle, and simultaneously starting to slide on it. In other words, the obstacle is moved aside by the drone pushing forward while the point of contact slides on the surface of the obstacle and on the robot's shell, resulting in the combined Push-and-Slide condition (mixed light blue

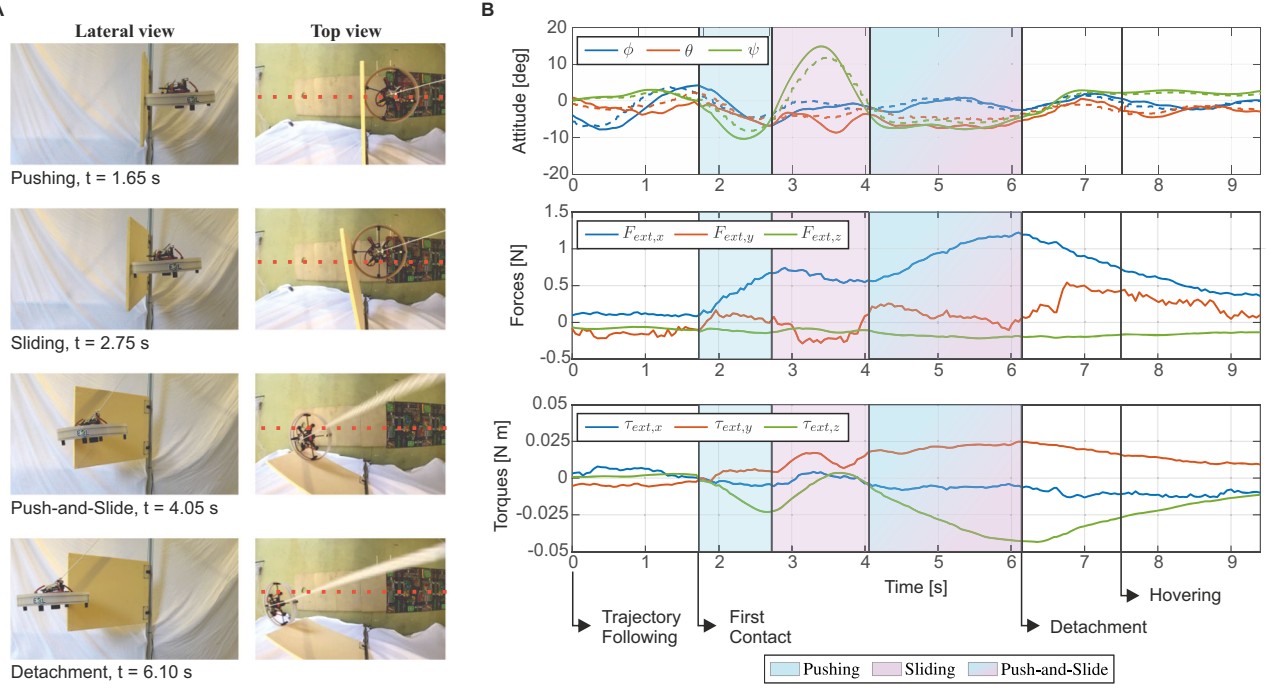

**Fig. 4 | Experiment with the mid value of stiffness, highlighting the phases of the traversal. A** Still frames from lateral (standard camera) and top (fisheye camera) views. Reference path depicted with a red dashed line. Direction of motion from right to left. **B** Attitude (roll $\phi$, pitch $\theta$, yaw $\psi$), external forces (longitudinal $F_{\mathrm{ext},x}$, lateral $F_{\mathrm{ext},y}$, vertical $F_{\mathrm{ext},z}$), and external torques (around the longitudinal axis $\tau_{\mathrm{ext},x}$, around the lateral axis $\tau_{\mathrm{ext},y}$, around the vertical axis $\tau_{\mathrm{ext},z}$) during the execution of the task. Commanded variables are reported with dotted lines.

and pink zone), which is indicated by increasing force on the longitudinal axis $F_{ext,x}$ and torque around the vertical one $\tau_{ext,z}$, while the yaw remains unaffected. Thereafter, the detachment is smooth and the drone keeps following the reference path, until it receives the hovering command.

We demonstrate the versatility and repeatability of the proposed strategy by performing ten experiments for each of the three stiffness values (Supplementary Movie S2). During all experiments, the reference speed is set to 0.15 m s⁻¹. The quantitative analysis in Fig. 5 refers only to the physical interaction, i.e., between the first contact and the detachment. The success rate (Fig. 5F) shows the number of successful tests out of ten. Since there is one failure for the highest stiffness, the rest of the metrics are evaluated on nine experiments to have the same number of data points for each stiffness. The five quantitative metrics for such analysis are explained in detail in the Methods chapter. The performance in terms of tracking the reference path is assessed by computing the errors in lateral position and longitudinal velocity. Statistical analysis using the Mann–Whitney $U$ test does not indicate a significant difference across different stiffness values ($p$ value > 0.05). As depicted in Fig. 5A, the drone does not experience drifts exceeding 0.2 m during the interaction. Although the data exhibits greater dispersion with increasing stiffness, the median values remain similar for all three stiffness levels, specifically at 0.084 m, 0.098 m, and 0.090 m, respectively. Figure 5B shows non-zero velocity errors, which are expected as the drone adjusts its speed during interaction with the environment. The medians of the velocity errors are similar across distributions, measuring 0.108 m s⁻¹, 0.104 m s⁻¹, and 0.109 m s⁻¹, respectively. The maximum force values along the longitudinal axis are depicted in Fig. 5C. Values increase with stiffness, reflecting the drone's need to push more to traverse stiffer obstacles. The median of the force is equal to 1.193 N, 1.272 N, and 1.670 N for the three values of stiffness. Statistical analysis reveals that the distributions of data, specifically for the low and high stiffness values, are statistically different from each other with a significance level of 95% (Mann–Whitney $U$ test, $p$ value < 0.05). The stability of the drone during interaction is assessed by analyzing the attitude oscillations (Fig. 5D, E). The amplitude of oscillations is calculated using the root mean square (RMS) of the roll angle $\phi$ and pitch angle $\theta$. Across the three stiffness values, the drone exhibits minimal oscillations, with medians well within the admissible roll and pitch angle limits [−20, 20] deg. Specifically, the medians are reported as 2.664 deg, 2.100 deg, and 3.525 deg for the roll and 3.935 deg, 5.057 deg, and 5.828 deg for the pitch, respectively. Statistical analysis for roll oscillations indicates no significant difference among the data distributions (Mann–Whitney $U$ test, $p$ value > 0.05). In contrast, the distributions of data for pitch oscillations show statistical differences for some stiffness values (Mann–Whitney $U$ test, *$p$ < 0.05 between low and mid, and **$p$ < 0.01 for low and high). This observation aligns with expectations, as the drone needs to pitch more to generate additional force when encountering obstacles with higher stiffness. The presented results imply that the performance of our approach in traversing compliant obstacles is consistent and independent of the obstacle compliance in the tested range, as further verified by the fact that the drone was capable of traversing in almost all the experiments (success rates over ten experiments reported in Fig. 5F).

**Ablation study.** The interaction versatility and adaptability demonstrated by the drone in the experiments arise from closely coupled body morphology and sensory driven control feedback. We conduct a series of experiments to illustrate that successful traversal of the obstacle cannot be achieved with either component alone (Fig. 6 and Supplementary Movie S3).

Initially, we adopt a squared cage and conduct experiments for each of the three values of stiffness. The non-streamlined shape of the cage leads the yaw rotation of the drone to become constrained with the rotation of the surface. Consequently, the drone remains stuck in the Pushing mode and could not slide or realign with the straight path (test for mid stiffness in Fig. 6A). It is worth noting that defining thresholds, as done in previous works, might facilitate the transition into a lateral Sliding mode. However, this approach contradicts our

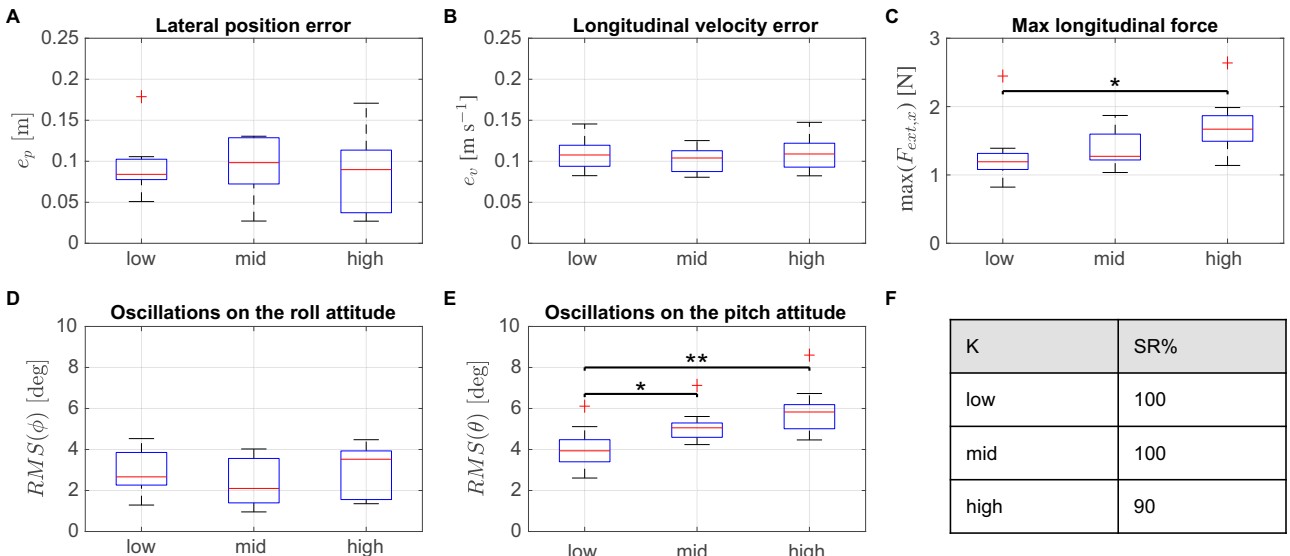

**Fig. 5 | Performance of the NMPC in terms of path following and stability during the interaction phase for different values of stiffness. A** tracking error of the lateral position; **B** longitudinal velocity; **C** max longitudinal force; **D** oscillations on the roll attitude; **E** oscillations on the pitch attitude. Mann–Whitney $U$ test performed for all possible combination of distribution; box plots indicate median (middle line), 25th, 75th percentile (box) and 5th and 95th percentile (whiskers) as well as outliers (single points); number of tests used for the statistical analysis = 9 for each stiffness (over a total of ten experiments each); $p$ > 0.05, *$p$ ≤ 0.05, **$p$ ≤ 0.01. **F** Success rate (SR %) of the proposed strategy for the different values of stiffness K.

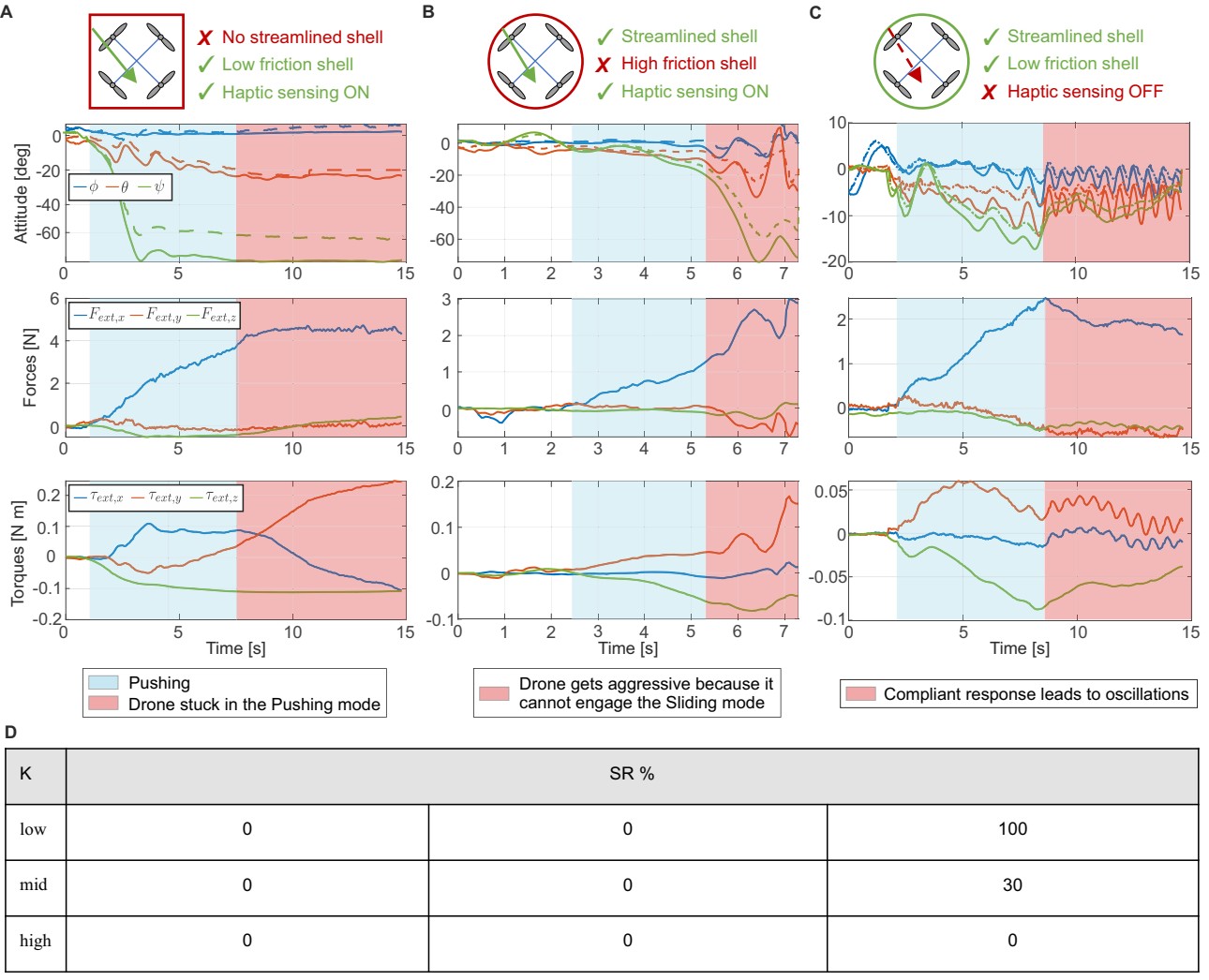

**Fig. 6 | Ablation study to validate the tight interplay between body morphology and sensory driven control feedback. A** Drone embodying a non-streamlined, squared cage. **B** Nominal cage with high-friction material on the outer shell. **C** Drone embodying the nominal cage and exploiting a controller without external wrench feedback and desired impedance behavior. **D** Success rate (%) of the three approaches for the different values of stiffness K. We stopped at five experiments when we had five consecutive successes/failures. We continued to ten experiments in case of a mix of successes and failures to have a more representative and accurate success rate (mid stiffness, haptic sensing OFF).

strategy, which aims to demonstrate that successful traversal can be achieved using optimized body morphologies without the need for empirically tuned thresholds and switching conditions.

Then, we incorporate non-slip material on the nominal, disc-shaped cage in order to increase the friction on the outer shell. In this case the drone attempts to slide, but this results in aggressive maneuvers due to the high friction that hinders the transition to the Sliding mode (test for mid stiffness in Fig. 6B). In both cases the traversal failed in 100% of the experiments (Fig. 6D). These two sets of experiments validate the importance of a streamlined and low-friction body morphology that allows for transitions between the interaction modes, which are necessary for a successful traversal.

Lastly, we conduct experiments to demonstrate the importance of haptic sensing to inform the feedback control loop, both to monitor the effects of the wrench on the drone's dynamics and for reacting to the interaction with the environment during traversal. To this end, we disable the haptic sensing, thereby removing the wrench measurements from the dynamics model and the impedance term in the NMPC formulation. This allows the controller to track the reference trajectory, but the interaction is treated only as a disturbance (test for mid stiffness in Fig. 6C). For low stiffness obstacles, trajectory tracking, coupled with the low-friction surface and streamlined shape of the shell, is sufficient for successful traversal due to the weak mechanical response of the environment (small wrenches involved and negligible oscillations). However, as the stiffness of the obstacle increases, the lack of haptic-driven control causes the drone to fail. For mid and high stiffness, the drone fails to traverse the obstacle (reaching instability) in 70% and 100% of the experiments respectively (success rate in Fig. 6D), due to the higher wrenches and notable oscillations. These experiments demonstrate that the drone cannot rely solely on morphology-enabled behaviors to successfully traverse stiffer obstacles. They confirm the crucial role of the closed-loop haptic feedback controller in sensing and dampening the oscillations induced by the compliant obstacles during the interaction.

**Single branch traversal.** Experiments are conducted using real branches (Fig. 7 and Supplementary Movie S4). The drone successfully traverses two branches—one branch without leaves (Fig. 7A) and the other one with small twigs and leaves (Fig. 7B)—both attached to a fixed structure at a single anchor point. The

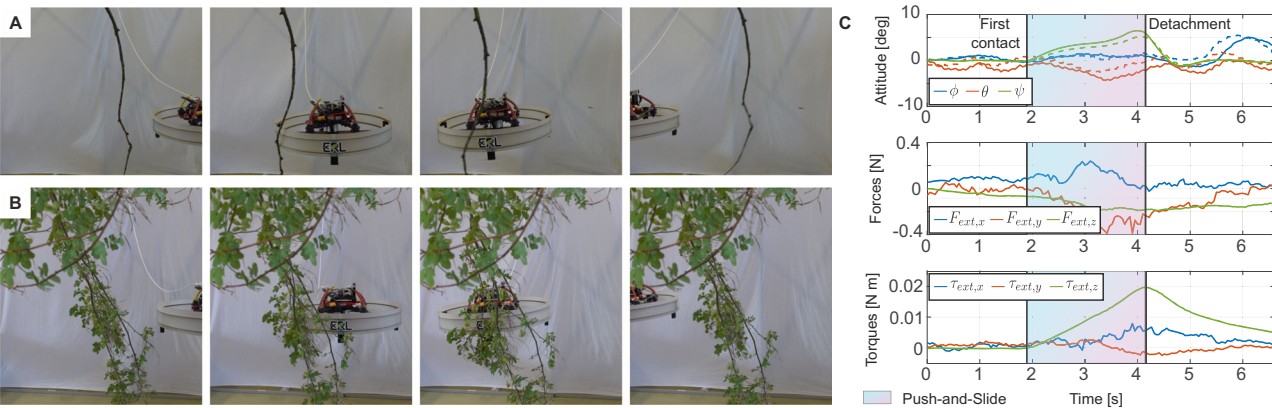

**Fig. 7 | Traversal of real vegetation.** Still frames from experiments with two different branches, one without leaves (**A**) and one with small twigs and foliage (**B**) (lateral view). **C** Attitude and external wrench during the traversal of the branch with foliage, with highlighted Push-and-Slide mode (colored region). Commanded variables are reported with dotted lines.

performance of the system is quantified using the evaluation metrics introduced in the previous analysis (reported in the "Methods" chapter). Table 1 highlights that errors in the lateral position remain close to zero across all experiments. The errors in longitudinal velocity, the longitudinal force, and the oscillations on the pitch angle, are smaller when compared to experiments with the foam plate. This reduction is attributed to the short interaction time and the rapid snapping away of the branches. The elongated shape and constraints of the branches results in a shorter and weaker interaction, leading to a quick Push-and-Slide mode. This is evident in Fig. 7C, illustrating the usage of Push-and-Slide mode due to increased longitudinal and lateral forces $F_{\text{ext},x}$ and $F_{\text{ext},y}$, and torque around the vertical axis $\tau_{\text{ext},z}$.

## Discussion

Surveying biotic and abiotic factors of Earth's ecosystems is essential to build the scientific knowledge needed to tackle the interconnected challenges of sustainability, climate change and declining biodiversity. Enabling drones to access cluttered vegetation would unlock the possibility to survey regions that cannot be sensed remotely by flying high in the sky. Drones designed for vegetation traversal hold the potential to revolutionize various crucial tasks, such as gathering environmental DNA in terrestrial ecosystems to conduct biodiversity surveys[3] or deploying sensors in trees and crops for environmental monitoring and precision agriculture[18,19,39].

In this work, we demonstrate that underactuated quadrotors can traverse a single obstacle with unknown and varying compliance through direct physical interaction. Our embodied APhI strategy harnesses the synergy between body morphology and haptic-based feedback control–both components are indispensable for successful traversal. The low-friction and streamlined shell grants the drone the ability to interact with and slide along obstacles with its entire body. A minimalist optimization-based controller commands the drone to fly along a straight trajectory and uses force feedback to effectively dampen oscillations of the environment, even in the presence of unknown and varying elastic responses. The versatility of our embodied APhI strategy is validated through a series of traversal experiments involving obstacles with compliance values spanning over an order of magnitude. Our approach demonstrates seamless handling of different interaction modes without the need for complex switching policies.

These results form a solid foundation for addressing the open research challenges needed for successfully traverse multiple compliant obstacles, as expected in the real world. Overcoming existing limitations requires a holistic advancement in the drone's morphology, sensing capacity, and intelligence.

In the current prototype, the discoid shell leaves the propellers vulnerable to collisions with branches, twigs, and leaves from both above and below the drone. To mitigate this, a spherical protective cage can be employed. For instance, the authors have successfully utilized a hemispherical cage to safeguard the drone's propellers during branch landings[3]. Additionally, an energy-absorbing cage has the potential to enhance resilience against collisions with stiff obstacles[40]. This enhancement could potentially relax the assumption of quasi-static motion, enabling navigation at higher speed.

The existing haptic sensing method measures the net wrench acting on the drone, which fails to discern the locations of individual contacts on the body and accurately measure local interaction forces. Preliminary simulation results indicate that this limits the robot's ability to navigate environments with multiple compliant obstacles (refer to Supplementary Method 7 and Supplementary Movie S6). In scenarios where the drone attempts to traverse both soft and rigid obstacles simultaneously, it may become stuck on rigid obstacles due to its inability to detect paths of lower stiffness with a higher likelihood of traversability. Addressing this challenge requires higher-resolution haptic sensing, achievable through electronic skins[41], visual haptic sensors or whisker arrays[42], to estimate the compliance and traversability of obstacles. When combined with high-level path planning, the drone can use this information to plan trajectories toward more traversable areas. This integration, together with the extension of the haptic controller from 2D to 3D, will be crucial to effectively manage multiple interactions and to deploy the drone in complex natural environments.

**Table 1 | Quantitative results of the experiments with real branches**

| Metrics | Test 1 | Test 2 | Test 3 | Test 4 |
|---|---|---|---|---|
| $e_p$ [m] | 0.066 | 0.063 | 0.044 | 0.033 |
| $e_v$ [m s$^{-1}$] | 0.104 | 0.044 | 0.078 | 0.069 |
| max($|F_{\text{ext},x,k}|$) [N] | 0.250 | 0.211 | 0.182 | 0.239 |
| RMS($\phi$) [deg] | 1.615 | 1.325 | 1.299 | 0.742 |
| RMS($\theta$) [deg] | 2.875 | 3.471 | 1.134 | 1.695 |

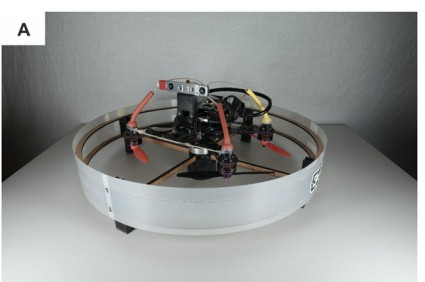
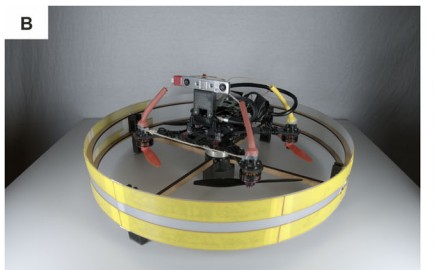
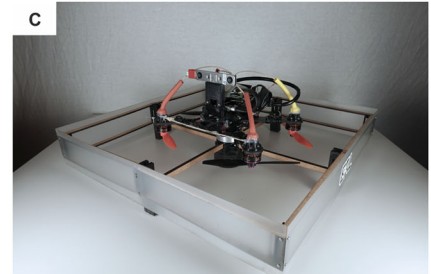

**Fig. 8 | Three cages used during the experiments. A** Streamlined, disc-shaped cage with low-friction shell. **B** Streamlined, disc-shaped cage with high-friction shell. **C** Non-streamlined, squared cage with low-friction shell.

## Methods

### Cages manufacturing

The frame of the shell was laser-cut (Trotec Speedy 360) from 2-mm medium-density fiberboard (MDF) panels. The components of the frame were connected via 3D-printed elements (Stratasys F120, ABS) and fixed with screws. The external surface of the shell was made of fiberglass (FR-40-HF, 0.2 mm) and connected to the frame via 3D-printed parts. The shell were designed such that only the external fiberglass surface can establish contact with the surroundings. Such a shell further shields the propellers and protects the inner components of the quadrotor. For the high-friction material, we laser cut strips of Dycem Non-Slip (Dycem Ltd). Figure 8 depicts the main cage, one with high-friction material on the shell, and the square variant.

### Quadrotor dynamics

In this work, we make use of a world, inertial frame $W$ with orthonormal basis $\{\hat{x}_W, \hat{y}_W, \hat{z}_W\}$ and a body frame $B$ with orthonormal basis $\{\hat{x}_B, \hat{y}_B, \hat{z}_B\}$ represented in world coordinates, fixed to the quadrotor with the origin coinciding with its center of mass (Fig. 2A).

The quadrotor dynamics are described through a rigid-body approach in which the forces resulting from each propeller are combined in the thrust force vector $\boldsymbol{T} = [0, 0, mc]^T$—where $c$ is the mass-normalized thrust—and into the torques $\boldsymbol{\tau}$ around the body axis, both terms expressed in body frame $B$. Following the rigid-body approach we can write the translational and rotational dynamics of the drone's CoM by deriving the state variables as follows—the dot ($\dot{}$) on the variables expresses their temporal derivative:

$$
\begin{aligned}
\dot{\boldsymbol{p}} &= \boldsymbol{v} \\
\dot{\boldsymbol{v}} &= -g\hat{\boldsymbol{z}}_W + \boldsymbol{R}_B^W(\boldsymbol{T} + \boldsymbol{F}_{\text{ext}})/m \\
\dot{\boldsymbol{q}} &= \boldsymbol{q} \odot [0 \; \boldsymbol{\omega}^T/2]^T \\
\dot{\boldsymbol{\omega}} &= \boldsymbol{J}^{-1}(\boldsymbol{\tau} - \boldsymbol{\omega} \times (\boldsymbol{J}\boldsymbol{\omega}) + \boldsymbol{\tau}_{\text{ext}})
\end{aligned}
\tag{1}
$$

The position and velocity of the quadrotor's CoM are $\boldsymbol{p} = [p_x, p_y, p_z]^T$ and $\boldsymbol{v} = [v_x, v_y, v_z]^T$, both expressed in the world frame $W$. The body rates $\boldsymbol{\omega} = [\omega_x, \omega_y, \omega_z]^T$, instead, are in body frame $B$. The quaternion $\boldsymbol{q} = [q_w, q_x, q_y, q_z]^T$ defines the orientation of the quadrotor; the rotation matrix that maps from body frame to world frame (function of $\boldsymbol{q}$) is denoted as $\boldsymbol{R}_B^W$, and can be also expressed with Euler angles $\phi, \theta, \psi$ (roll, pitch, yaw). The symbol $\odot$ denotes the quaternion multiplication operator. Further, $m$ is the mass of the platform, $g = 9.81 \, \text{m s}^{-2}$ is the gravitational acceleration, along the opposite direction of $\hat{\boldsymbol{z}}_W$, and $\boldsymbol{J} = \text{diag}(J_x, J_y, J_z)$ is the inertia matrix. Finally, $\boldsymbol{F}_{\text{ext}}$ and $\boldsymbol{\tau}_{\text{ext}}$ are respectively the external force and torque acting on the drone during the interaction—the external wrench $\boldsymbol{w}_{\text{ext}}$ acting on the CoM is defined as $[\boldsymbol{F}_{\text{ext}}^T \, \boldsymbol{\tau}_{\text{ext}}^T]^T \in \mathbb{R}^6$.

### NMPC formulation

For the NMPC problem we define the state $\boldsymbol{x}$ and the input $\boldsymbol{u}$ of the system as follows:

$$
\begin{aligned}
\boldsymbol{x} &= [\boldsymbol{p}^T \boldsymbol{v}^T \boldsymbol{q}^T \boldsymbol{\omega}^T]^T \in \mathcal{X} \in \mathbb{R}^{13} \\
\boldsymbol{u} &= [c \, \boldsymbol{\tau}^T]^T \in \mathcal{U} \in \mathbb{R}^4
\end{aligned}
\tag{2}
$$

Then, the discrete dynamics of the quadrotor $\boldsymbol{x}_{k+1} = \boldsymbol{f}(\boldsymbol{x}_k, \boldsymbol{u}_k, \boldsymbol{w}_{\text{ext},k})$ are obtained by discretizing the equations of motion with an explicit Runge–Kutta method of the 4th order. We define a quadratic optimization problem using a multishooting scheme and solve the following discretized nonlinear optimal control problem (OCP):

$$
\begin{aligned}
\min_{\boldsymbol{u}_k} \; & \| \boldsymbol{h}_N(\boldsymbol{x}_N) \|_{Q_N}^2 + \sum_{k=0}^{N-1} \left( \| \boldsymbol{h}_x(\boldsymbol{x}_k) \|_{Q_x}^2 + \| \boldsymbol{h}_u(\boldsymbol{u}_k) \|_{Q_u}^2 \right) \\
\text{s.t.} \quad & \boldsymbol{x}_{k+1} = \boldsymbol{f}(\boldsymbol{x}_k, \boldsymbol{u}_k, \boldsymbol{w}_{\text{ext},k}) \quad k = 0, \dots, N-1 \\
& \boldsymbol{x}_0 = \boldsymbol{x}_{\text{init}}, \quad \boldsymbol{x}_{\min} \le \boldsymbol{x}_k \le \boldsymbol{x}_{\max} \\
& \boldsymbol{u}_{\min} \le \boldsymbol{u}_k \le \boldsymbol{u}_{\max}
\end{aligned}
\tag{3}
$$

as a sequential quadratic program (SQP), executed in a real-time iteration scheme[43], with a receding-horizon—we discretize the system evolution into $N$ steps over a time horizon $t$, and $k$ is the current time step. The goal cost $\boldsymbol{h}_N(\boldsymbol{x}_N)$, the tracking cost $\boldsymbol{h}_x(\boldsymbol{x}_k)$, and the action regularization cost $\boldsymbol{h}_u(\boldsymbol{u}_k)$ refer to the objective cost vectors of the optimal control problem, which is implemented using the open source ACADO toolkit[44].

**Constraints.** In the OCP, we define constraints on some components of the input and state vectors, specifically on the mass-normalized thrust and the orientation, applied at each time step $k$:

$$
\begin{aligned}
0 &\le c_k \le c_{\max} \\
-\phi_{\min} &\le \phi_k \le \phi_{\max} \\
-\theta_{\min} &\le \theta_k \le \theta_{\max}
\end{aligned}
\tag{4}
$$

where the roll and pitch angles $\phi_k$ and $\theta_k$ are converted from the quaternion—via a direct trigonometric formula—before performing the optimization. Whereas constraining the thrust depends on physical limitations due to the hardware, we artificially constrain the orientation—in our case the roll and pitch angles—to limit the forces and torques exchanged with the environment. Since the interaction task is defined in the quadrotor CoM, we have to consider that its underactuation prevents direct control of body wrenches, thus imposing the drone to tilt in order to apply forces and torques on the plane orthogonal to the vertical axis (i.e., the push direction and the sliding plane). Having constraints on the angles allow us to ensure stable flight during the interaction, which, in combination with low velocities, ensures small deviations from the translational equilibrium state. For theoretical

guarantees on the stability of underactuated drones during physical interaction, we refer the readers to ref. 45.

**Objective cost vectors and weights.** In the objective cost, we distinguish three cost vectors:

$$
\boldsymbol{h}_x(\boldsymbol{x}_k) = \begin{bmatrix} p_{y,k} - p_{y,\mathrm{ref}} \\ p_{z,k} - p_{z,\mathrm{ref}} \\ \boldsymbol{v}_k - \boldsymbol{v}_{\mathrm{ref}} \\ \dot{\boldsymbol{v}}_k - \dot{\boldsymbol{v}}_{\mathrm{imp}} \\ \psi_k - \psi_{\mathrm{ref}} \\ \boldsymbol{\omega}_k - \boldsymbol{\omega}_{\mathrm{ref}} \end{bmatrix} \in \mathbb{R}^{12}
$$

$$
\boldsymbol{h}_u(\boldsymbol{u}_k) = \begin{bmatrix} c - c_{\mathrm{ref}} \\ \boldsymbol{\tau} - \boldsymbol{\tau}_{\mathrm{ref}} \end{bmatrix} \in \mathbb{R}^{4} \tag{5}
$$

$$
\boldsymbol{h}_N(\boldsymbol{x}_N) = \begin{bmatrix} p_{y,N} - p_{y,\mathrm{ref}} \\ p_{z,N} - p_{z,\mathrm{ref}} \\ v_{x,N} - v_{x,\mathrm{ref}} \end{bmatrix} \in \mathbb{R}^{3}
$$

where we denote quantities related to the reference path with subscript "ref" and quantities computed by the internal impedance module with subscript "imp".

The first two vectors $\boldsymbol{h}_x(\boldsymbol{x}_k)$ and $\boldsymbol{h}_u(\boldsymbol{u}_k)$ keep track of the error (for a subset of the state and for the input) and the interaction, for each step of the receding horizon. They consist of a position term, only on the lateral and vertical directions, to prevent the drone from drifting from the path, and a velocity term for progression, assigned to all directions to keep a longitudinal velocity along the path and zero velocity on the other two directions. Another term takes into account the mismatch between the quadrotor dynamics and desired impedance dynamics, which is needed to handle a safe physical interaction. A yaw term is used to keep the platform at a fixed yaw angle (aligned with the path), and finally the angular velocity, thrust, and torque terms act as regularization terms (reference thrust is set to hovering value, and angular velocity and torques set to zero, valid for both free-flight and interaction). To follow a straight path, our choice of assigning a reference velocity instead of a reference position (on the longitudinal axis) simplifies the implementation, as the reference velocity can be set at the beginning and kept fixed, whereas a reference position would need to be updated at each iteration as the drone moves forward. The last cost vector $\boldsymbol{h}_N(\boldsymbol{x}_N)$ acts as a terminal cost since it only depends on the terminal state. We can distinguish a position term needed for path following (two directions) and a velocity term (only along the reference path direction) needed for progression.

The cost vectors' weighted norms are multiplied by the matrices $\boldsymbol{Q}_x \in \mathbb{R}_{\geq 0}^{12 \times 12}, \boldsymbol{Q}_u \in \mathbb{R}_{>0}^{4 \times 4}$, and $\boldsymbol{Q}_N \in \mathbb{R}_{\geq 0}^{3 \times 3}$, which are constant throughout the prediction horizon. Regarding the velocity term in the matrix $\boldsymbol{Q}_x$, different coefficients can be used for the velocity along the path following direction ($Q_{v_x}$) and the remaining two directions ($Q_{v_{y,z}}$). The terminal weights of $\boldsymbol{Q}_N$ have been tuned and chosen by performing physics-based simulations in the Gazebo 3D simulator. Considering the tracking of the reference path as well as a safe, quasi-static interaction as performance metrics, we tuned and selected the weights that allowed us to generalize the robot performance over different values stiffness.

**Desired impedance behavior for physical interaction.** The desired impedance acceleration is computed as the one of a rigid body with a specific impedance—mass, damping, and stiffness—subjected to an external force acting on it. In such a way, we can reshape the apparent mechanical characteristics of the robot:

$$
\dot{\boldsymbol{v}}_{\mathrm{imp}} = \boldsymbol{M}^{-1}\big(-\boldsymbol{D}(\boldsymbol{v} - \boldsymbol{v}_{\mathrm{ref}}) - \boldsymbol{K}(\boldsymbol{p} - \boldsymbol{p}_{\mathrm{ref}}) + \boldsymbol{F}_{\mathrm{ext}}\big) \tag{6}
$$

where the three positive-definite diagonal matrices $\boldsymbol{M}, \boldsymbol{D}$ and $\boldsymbol{K} \in \mathbb{R}_{\geq 0}^{3 \times 3}$ define the apparent mass, damping, and stiffness of the desired dynamics of the vehicle.

We selected the mass of the drone (1.2 kg) as a suitable value for the diagonal coefficients of $\boldsymbol{M}$. Then, in order to achieve a safe and smooth behavior, as well as to properly dampen eventual oscillations provoked by the elastic response of the obstacles, we select the damping coefficients of $\boldsymbol{D}$ equal to 1 to have the same order of magnitude of the apparent mass. We select $\boldsymbol{K} = \boldsymbol{O}^{3 \times 3}$ and the external force only on the $y$–$z$ axis and to be positive. This choice is useful because: (1) we predominantly want a damping behavior during the interaction, discarding additional elastic behavior of the system in response to the interaction, and (2) we want the drone to push against the obstacle in the direction opposite of the interaction force. In such a way, similarly to relevant work in human-robot interaction[46], the robot can indirectly be guided toward the direction of traversal when the obstacle moves aside. Further, as highlighted in ref. 47, for unstructured environments it is safer to have low values of apparent stiffness when the reference speed is low.

## Statistical analysis

The statistical analysis reported in the "Results" chapter quantifies the performance and repeatability of our solution. To obtain the distributions of data that we used for the statistical analysis of the performance for varying stiffnesses, we only consider the interval of time when interaction occurs (first contact to detachment) for each experiment. We define this interval as $[1, M]$ with $M$ equal to the number of samples during the interaction, and extract the following metrics:

- The tracking error of lateral position over the interaction interval, by computing the mean absolute error (MAE): $e_p = \frac{1}{M}\sum_{k=1}^{M} |p_{y,k} - p_{y,\mathrm{ref}}|$. This metric describes how close the drone is to the straight reference path.
- The tracking error of longitudinal velocity during the interaction interval, using the MAE as above: $e_v = \frac{1}{M}\sum_{k=1}^{M} |v_{v,k} - v_{v,\mathrm{ref}}|$. This metric is expected to deviate from zero during the interaction, due to the response of the environment impeding the drone's motion.
- The interaction force along the longitudinal axis, by saving the maximum value during the interaction interval: $\max(|F_{\mathrm{ext}_x,k}|)$ with $k = [1, M]$. This metric is connected to stability as it tracks high forces that may lead to undesirable behavior.
- The amplitude of the oscillations occurring on the attitude of the drone; we define this quantity by computing the RMS (root mean square) for both roll $\phi$ and pitch $\theta$: $\mathrm{RMS}(\phi) = \sqrt{(\sum_{k=1}^{M} \phi_k^2)/M}, \mathrm{RMS}(\theta) = \sqrt{(\sum_{k=1}^{M} \theta_k^2)/M}$. This metric also refers to stability as it shows whether the drone is subject to abrupt oscillations on the attitude during the interaction.

Thus, for $N$ experiments, we have a distribution composed of $N$ values for the introduced metrics. The Mann–Whitney $U$ test was performed in MATLAB R2021a (MathWorks, MA, USA). The "boxplot" function is used to create box plots, which provides median, 25th and 75th percentiles (which are also called first quartile Q1 and third quartile Q3), 5th and 95th percentile, and outliers. The latter are computed in a box plot based on the interquartile range (IQR), which is the difference between the Q3 and Q1 of the dataset. The lower bound for detecting outliers is calculated as $Q1 - 1.5*IQR$, whereas the upper bound for detecting outliers is calculated as $Q3 + 1.5*IQR$. However, it is important to note that the factor of 1.5 used in calculating the outlier thresholds can be adjusted based on specific needs or domain-specific standards. Finally, the "ranksum" function is used to compute the $p$ values.

## Data availability

The datasets generated and analyzed in this study, as well as the scripts for the analysis and the CAD files, have been deposited in the ETH Research Collection database, available for open access https://doi.org/10.3929/ethz-b-000662212. Source data are provided with this paper.

## Code availability

The controller code and the simulation workspace are available on Zenodo at https://doi.org/10.5281/zenodo.10798275 for open access.

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

## Acknowledgements

This work was supported by the Swiss National Science Foundation through the Eccellenza under Grant PCEFP2_186865.

## Author contributions

E.A. and S.M. conceived the project. E.A., C.G. and S.M. conceptualized the idea and investigated the methodology. E.A. and S.M. conceptualized, designed and manufactured the robot's body. D.M. implemented the controller in simulation and evaluated the performance. E.A., C.G. and D.M. transferred the controller on the real robot. E.A. and C.G. conducted the experiments. E.A. performed the ablation studies. E.A. performed the statistical data analysis. L.P. and S.M. provided insights throughout the project. E.A. and S.M. prepared the manuscript, and all the authors provided feedback during subsequent revisions. S.M. administered the research and provided funding and principal supervision.

## Competing interests

The authors declare no competing interests.
