## [Peer Review File · Nature Communications]

Synergistic morphology and feedback control for traversal of unknown compliant obstacles with aerial robotsREVIEWER COMMENTS

Reviewer #1 (Remarks to the Author):

Inspired by the physical interactions and traversal behaviors of animals in complex vegetation, the manuscript titled "Synergistic Morphology and Feedback Control for Traversal of Unknown Compliant Obstacles with Aerial Robots" explores the challenge of aerial robots traversing unknown compliant obstacles by pushing and sliding. Specifically, the study begins by mimicking the streamlined shape and acute sensory abilities found in most insects, designing the aerial robot with a disc-shaped shell and equipping it with a force/torque sensor to enable the aerial robot to establish and perceive contact at any position on the shell, facilitating sliding along obstacles. Subsequently, using the Nonlinear Model Predictive Control (NMPC) method, the aerial robot achieves both traversal of flexible obstacles and following a straight path simultaneously. Notably, the control strategy devised in this study eliminates the need for high-level switching conditions between pushing and sliding; instead, it computes the desired impedance acceleration through the internal impedance module to facilitate physical interaction with the flexible obstacles. This approach employs proportional-derivative control to appropriately suppress oscillations caused by the elastic response of the obstacles. The feasibility of the proposed solution is extensively validated through numerous experiments, and various ablative experiments demonstrate the necessity of each component in the design.

In summary, this article addresses the constraints associated with interactions between aerial robots and compliant obstacles, presenting an intriguing solution where aerial robots emulate animals by pushing and sliding to traverse such obstacles. The content of the article is engaging, and the experimental setup and results are reasonable. While the study primarily considers interactions in a two-dimensional plane and low-speed flight, the concept presented in the article is novel and should inspire further research. Here are my comments:

1. In the article, aerial robots traverse compliant obstacles by pushing and sliding. However, in real jungle scenarios with dense foliage, there is a possibility of the aerial robot getting stuck. How should such situations be addressed?
2. The article focuses on interactions in a two-dimensional plane, but sometimes it may not be optimal to limit traversal to two dimensions, especially when obstacles have a significant proportion of their length in the y-axis and z-axis directions.
3. The study considers low-speed movement of aerial robots (with a reference speed of 0.15 m/s). Given the need for contact with obstacles, increasing speed may introduce significant momentum and make traversing obstacles highly challenging.
4. Differentiating and recognizing various types of obstacles becomes crucial for aerial robots. It is important to develop a method for identifying and categorizing obstacles to determine whether direct avoidance or the traversal method proposed in the article is more appropriate in real-world scenarios.
5. There is a grammatical error in line 184: "which is can be seen." It should be revised to maintain proper grammar.

Reviewer #2 (Remarks to the Author):

This paper introduces a collision-resilient quadrotor which can traverse unknown compliant obstacles. This is achieved by studying different shapes of the cage design as well as a nonlinear model predictive controller (NMPC) design. The NMPC incorporates objectives and constraints to track the desired path and ensure safe interactions. Experimental results are provided with three different cage designs and three stiffness of the obstacles. Overall, this paper is well written, and the attached videos are of high quality, which helps explain the results of this work. However, the reviewer has some major concerns about this work as explained below.

1. Introduction: this paper is well motivated, but the literature review needs to be extended. For example, the impedance/admittance control, pushing, and sliding motion patterns, NMPC for drone-environment physical interaction have all been introduced in existing work
a. Patnaik, K., Mishra, S., Sorkhabadi, S. M. R., & Zhang, W. (2020, October). Design and Control

of SQUEEZE: A Spring-augmented QUadrotor for intEractions with the Environment to squeeZE-and-fly. In 2020 IEEE/RSJ International Conference on Intelligent Robots and Systems (IROS) (pp. 1364-1370). IEEE.

b. Khedekar, N., Mascari, F., Papachristos, C., Dang, T., & Alexis, K. (2019, May). Contact-based navigation path planning for aerial robots. In 2019 International Conference on Robotics and Automation (ICRA) (pp. 4161-4167). IEEE.

c. Lee, D., Seo, H., Kim, D., & Kim, H. J. (2020, May). Aerial manipulation using model predictive control for opening a hinged door. In 2020 IEEE International Conference on Robotics and Automation (ICRA) (pp. 1237-1242). IEEE.

The authors should make clear the technical novelty of their work.

2. NMPC design: how the reference force and torque are determined in (5)? How is the horizon N determined in the NMPC design, especially given the unknown obstacles? It will also be important to perform some sort of ablation study on the impact of NMPC parameters to the performance of the overall system. How sensitive is the controller to the accuracy of wrench measurement and system modeling? Lastly, as the authors mentioned in the paper, when the drone starts and exits physical contact with the environment, there is a jump in the external force which may cause instability. How does the NMPC handle this stability issue?

3. The proposed controller relies on the integrated force/torque sensor to understand the interaction with the environment – this means the contact must take place in the designated area where sensor measurement can be taken. This may not be the case for traversing through forests with branches and leaves. Some discussions should be included to discuss this limitation.

4. I notice that there is a cable connected to the drone in all experimental videos. Is that cable for data communication or power? It introduces some concerns that the cable may either obstruct the motion of the drone or provide stability support to the drone under disturbance. The authors need to ensure that there is no force at any time during the experiment.

5. How do you determine the three different values of stiffness in experiments? This may have major impacts on the results shown in Fig 5 and Fig 6 so this point should be made clear.

6. For the non-streamlined design in the experiments, I understand that it does not work because it cannot slide. However, it seems that it can just push the obstacle away if the drone follows the original path (instead of maintaining contact to the obstacle). I guess it is just because this design does not fit the controller design, so the current conclusion can be a bit misleading.

7. In Figure 6, the figure has the haptic feedback off for C. I am confused how the controller will work without the force/torque sensor measurement. Some further explanations may be helpful.

8. The result shown in Figure 7 is with only one branch of leaves – what will happen if there are multiple contacts simultaneously? It seems that the force/torque sensor will provide a net wrench and it might confuse the controller. Some additional discussions and experimental results will be needed to make this point clear.

Reviewer #3 (Remarks to the Author):

This paper presents an approach for aerial robots to traverse unknown compliant obstacles using direct physical interaction. The combination of the task-oriented design and haptic feedback control shows promise for improving aerial robots' ability to move through dense vegetation. I have a few main comments:

- The introduction motivates the problem well, but is a bit lengthy. I would suggest tightening up the introduction to more clearly highlight the key gaps this work aims to address.
- In the results, it would be good to provide some quantitative metrics on the tracking performance for different stiffness values, rather than just the qualitative analysis. This would better showcase the versatility of the approach across varying compliance.
- The ablation study clearly demonstrates the importance of both the morphology and haptic feedback control. However, it is unclear why the controller alone sometimes succeeds for low stiffness but always fails for higher stiffness. Some discussion on the limitations of the controller by itself would strengthen this ablation study.

- The real branch experiments provide a nice proof-of-concept, but lack the detailed analysis done for the hinged plate experiments. It would be good to provide more quantification of the modes of interaction and forces/torques for the branch experiments too.

- The writing could be tightened up in some areas - there is occasional verbose language that makes the paper longer than necessary.

Response to Reviewers

Synergistic Morphology and Feedback Control for Traversal of Unknown Compliant Obstacles with Aerial Robots

Emanuele Aucone, Christian Geckeler, Daniele Morra, Lucia Pallottino, and Stefano Mintchev

Dear Reviewers,

We would like to express our deep gratitude for the valuable feedback and reviews, which helped us to greatly improve our manuscript. We also want to thank you for the recognition of our work. The manuscript has been revised according to the suggested reviews, the major changes are summarized here, and detailed point-by-point revisions follow below.

Main revisions:

- The Introduction chapter has been extended with additional related works.
- The Results chapter has been extended with further clarifications and additional results to better quantify the experimental validation conducted on the hinged compliant surface and branches.
- The Discussion chapter has been restructured with clarifications and further information about the limitations and the future investigations of our work.
- An ablation study for the NMPC parameters, further explanations about the controller implementation, and additional experiments with multiple obstacles have been added to the Supplementary Information.
- Writing has been revised to avoid repetitions and verbose language.

In addition, formatting, figures, references, and supplementary information have been adapted according to the checklists and the guidelines, and the zip file for code/software has been updated with the new data, scripts, and ReadMe for the additional analysis performed.

Thank you very much once again, and we hope the revision will meet your expectations.

Best regards,

Emanuele Aucone, Christian Geckeler, Daniele Morra, Lucia Pallottino, and Stefano Mintchev

Reviewer #1

Inspired by the physical interactions and traversal behaviors of animals in complex vegetation, the manuscript titled "Synergistic Morphology and Feedback Control for Traversal of Unknown Compliant Obstacles with Aerial Robots" explores the challenge of aerial robots traversing unknown compliant obstacles by pushing and sliding. Specifically, the study begins by mimicking the streamlined shape and acute sensory abilities found in most insects, designing the aerial robot with a disc-shaped shell and equipping it with a force/torque sensor to enable the aerial robot to establish and perceive contact at any position on the shell, facilitating sliding along obstacles. Subsequently, using the Nonlinear Model Predictive Control (NMPC) method, the aerial robot achieves both traversal of flexible obstacles and following a straight path simultaneously. Notably, the control strategy devised in this study eliminates the need for high-level switching conditions between pushing and sliding; instead, it computes the desired impedance acceleration through the internal impedance module to facilitate physical interaction with the flexible obstacles. This approach employs proportional-derivative control to appropriately suppress oscillations caused by the elastic response of the obstacles. The feasibility of the proposed solution is extensively validated through numerous experiments, and various ablative experiments demonstrate the necessity of each component in the design. In summary, this article addresses the constraints associated with interactions between aerial robots and compliant obstacles, presenting an intriguing solution where aerial robots emulate animals by pushing and sliding to traverse such obstacles. The content of the article is engaging, and the experimental setup and results are reasonable. While the study primarily considers interactions in a two-dimensional plane and low-speed flight, the concept presented in the article is novel and should inspire further research.

Response:

Thank you very much for the review and the recognition of this work. The revisions have been made according to the specific comments below. Comments 1, 2 and 4 have been answered singularly whereas the revisions have been grouped as they all refer to the "Discussion" chapter, which has been extended according to these comments and some other reviewers (e.g. interaction with multiple obstacles).

In the following we report the main revisions in the manuscript. The direct answers to the reviewers' comments are presented in **blue font**, whereas the newly added content to the Main article is presented in **red font** and the removed content is also **strikethrough**.

Here are my comments:

1. In the article, aerial robots traverse compliant obstacles by pushing and sliding. However, in real jungle scenarios with dense foliage, there is a possibility of the aerial robot getting stuck. How should such situations be addressed?

Response:

The domain of APhI with compliant obstacles is a novel area of research. This article represents the initial exploration of this domain, focusing on the interaction with single obstacles. As the reviewer correctly pointed out, the risk of the drone becoming entangled in vegetation increases as the environment becomes denser. We believe that this challenge can be addressed by synergistically enhancing both the robot's physical structure (body) and its decision-making capabilities (brain).

The current prototype features a discoid shell conceived to provide protection during interactions with the compliant plate used in the experiments. However, this design leaves the propellers exposed to slender obstacles that could collide with them and impede their rotation. To address this vulnerability, we suggest the incorporation of a protective cage or shell around the entire drone. In a related study by

the authors (see [3]), we demonstrated that a hemispherical cage offers effective protection against branch intrusion, allowing a drone to safely interact and land on branches with leaves and twigs.

Another essential step in reducing the risk of getting stuck is enhancing the drone's decision-making abilities. In our current work, the drone attempts to navigate through compliant obstacles following a straight trajectory, lacking the capacity to replan it based on the properties of the environment. Enabling the drone to assess the traversability of the surroundings would enable it to plan trajectories through compliant regions, thus minimizing the risk of getting stuck in regions with obstacles that cannot be traversed. This would require refined haptic sensing capabilities and novel planning strategies.

We have expanded the "Discussion" chapter to include these insights, which offer specific recommendations for future research directions.

[3] Aucone, E. et al. Drone-assisted collection of environmental DNA from tree branches for biodiversity monitoring. *Science Robotics* 8, eadd5762 (2023)

2. The article focuses on interactions in a two-dimensional plane, but sometimes it may not be optimal to limit traversal to two dimensions, especially when obstacles have a significant proportion of their length in the y-axis and z-axis directions.

Response:

The extension of interaction to three-dimensionality, allowing the drone to travel complex paths and interact anywhere around its' body, is an important future research step. This will require integrating a cage that can offer 3D protection and sensing to the drone, and extending the control algorithms to 3D trajectories.

We have discussed these two points in the "Discussion" chapter, where we present the limitations of current solutions and ideas to overcome them.

4. Differentiating and recognizing various types of obstacles becomes crucial for aerial robots. It is important to develop a method for identifying and categorizing obstacles to determine whether direct avoidance or the traversal method proposed in the article is more appropriate in real-world scenarios.

Response:

This topic extends the answer to the first comment, where we discussed future developments towards novel haptic-based navigation strategies. In the current prototype, haptic feedback is used to stabilize the drone, but it is possible to expand the scope to estimate the compliance of the surrounding obstacles. Such measures could provide drones with the ability to identify traversable areas. Consequently, novel planning strategies could allow drones to navigate through regions with higher compliance or directly avoid potentially dangerous interaction with very stiff obstacles.

The "Discussion" chapter has been extended accordingly.

Revision:

P. 7, Line 348:

~~This work also sets the stage for future developments aimed at overcoming current limitations. The drone now traverses compliant obstacles with a quasi-static motion to assure safe exploitation and transition between the interaction modes. For instance, if an abrupt first contact occurs the drone may not dampen out the high value of the external wrench upon impact, resulting in an aggressive pushing on the hinged surface. This effect could be mitigated by acting on the sensing level (e.g. filtering more the wrench measures or discarding abrupt spikes in the signal) or on the morphology by integrating dampening materials in the shell. Moreover, to interact with multiple obstacles simultaneously, higher~~

~~resolution and precise haptic sensing are essential, which can be achieved through electronic skins [31] or whisker arrays [32]. Our experiments emphasize the significance of tightly integrating body and mind to enhance aerial robots' performance when interacting with complex environments. Looking ahead, we envision the co-evolution of body morphologies, accurate sensory apparatus, and intelligent control systems playing a pivotal role in translating our current discoveries into practical outdoor applications.~~

These results form a solid foundation for addressing the open research challenges needed for successfully traverse multiple compliant obstacles, as expected in the real world. Overcoming existing limitations requires a holistic advancement in the drone's morphology, sensing capacity, and intelligence.

In the current prototype, the discoid shell leaves the propellers vulnerable to collisions with branches, twigs, and leaves from both above and below the drone. To mitigate this, a spherical protective cage can be employed. For instance, the authors have successfully utilized a hemispherical cage to safeguard the drone's propellers during branch landings [3]. Additionally, an energy-absorbing cage has the potential to enhance resilience against collisions with stiff obstacles [R1]. This enhancement could potentially relax the assumption of quasi-static motion, enabling navigation at higher speed.

The existing haptic sensing method measures the net wrench acting on the drone, which fails to discern the locations of individual contacts on the body and accurately measure local interaction forces. Preliminary simulation results indicate that this limits the robot's ability to navigate environments with multiple compliant obstacles (refer to Supplementary Information and Supplementary Movie S6). In scenarios where the drone attempts to traverse both soft and rigid obstacles simultaneously, it may become stuck on rigid obstacles due to its inability to detect paths of lower stiffness with a higher likelihood of traversability. Addressing this challenge requires higher-resolution haptic sensing, achievable through electronic skins [41], visual haptic sensors or whisker arrays [42], to estimate the compliance and traversability of obstacles. When combined with high-level path planning, drones could use this information to plan trajectories towards more traversable areas. This integration, together with the extension of the haptic controller from 2D to 3D, will be crucial to effectively manage multiple interactions and to deploy the drone in complex natural environments.

New references:

R1: Pooya Sareh et al., Rorigami: A rotary origami protective system for robotic rotorcraft. *Sci. Robot.* 3, eaah5228(2018). DOI:10.1126/scirobotics.aah5228

3. The study considers low-speed movement of aerial robots (with a reference speed of 0.15 m/s). Given the need for contact with obstacles, increasing speed may introduce significant momentum and make traversing obstacles highly challenging.

Response:

Thank you for pointing it out, the quasi-static motion is indeed essential in our strategy. We have made this clearer in the “Controller Rationale” section. We have also suggested in the Discussion chapter that an energy-absorbing cage has the potential to enhance resilience against collisions with stiff obstacles.

Revision:

P. 4, Line 163:

Finally, a set of safety constraints in the NMPC ~~creates~~ guarantees to avoid ~~bouncing conditions~~oscillations upon contact and detachment, as well as to permit a smooth and direct transition between the interaction modes thanks to the body shape. ~~In detail,~~ ~~E~~enforcing constraints on the dynamics of the drone (e.g. max pitch and roll angles), in combination with flying at low reference

speed, allows us to assure a quasi-static motion and a safer operation, by limiting the external wrench exchanged during interaction. Increasing speed would result in higher wrenches exerted on the drone, which would make the traversal more challenging (e.g. difficulty in smoothly transitioning between interaction modes due to the high speed and short interaction time) or destabilize the drone (e.g. drone not capable of counteracting a high wrench due to physical limitation of the platform).

5. There is a grammatical error in line 184: "which is can be seen." It should be revised to maintain proper grammar.

Response:

We have addressed the grammatical error.

Revision:

P. 4, Line 212:

... which ~~is~~ can be seen by the change in the force ...

Reviewer #2

This paper introduces a collision-resilient quadrotor which can traverse unknown compliant obstacles. This is achieved by studying different shapes of the cage design as well as a nonlinear model predictive controller (NMPC) design. The NMPC incorporates objectives and constraints to track the desired path and ensure safe interactions. Experimental results are provided with three different cage designs and three stiffness of the obstacles. Overall, this paper is well written, and the attached videos are of high quality, which helps explain the results of this work. However, the reviewer has some major concerns about this work as explained below.

Response:

Thank you very much for the review and the recognition of this work. The revisions have been made according to the specific comments.

In the following we report the main revisions in the manuscript. The direct answers to the reviewers' comments are presented in **blue font**, whereas the newly added content to the Main article is presented in **red font** and the removed content is also **strikethrough**.

1. Introduction: this paper is well motivated, but the literature review needs to be extended. For example, the impedance/admittance control, pushing, and sliding motion patterns, NMPC for drone-environment physical interaction have all been introduced in existing work. A) Patnaik, K., Mishra, S., Sorkhabadi, S. M. R., & Zhang, W. (2020, October). Design and Control of SQUEEZE: A Spring-augmented QUadrotor for intERactions with the Environment to squeeZE-and-fly. In 2020 IEEE/RSJ International Conference on Intelligent Robots and Systems (IROS) (pp. 1364-1370). IEEE. B) Khedekar, N., Mascarich, F., Papachristos, C., Dang, T., & Alexis, K. (2019, May). Contact-based navigation path planning for aerial robots. In 2019 International Conference on Robotics and Automation (ICRA) (pp. 4161-4167). IEEE. C) Lee, D., Seo, H., Kim, D., & Kim, H. J. (2020, May). Aerial manipulation using model predictive control for opening a hinged door. In 2020 IEEE International Conference on Robotics and Automation (ICRA) (pp. 1237-1242). IEEE. The authors should make clear the technical novelty of their work.

Response:

We have extended the related works to clarify the novelty of our work. The scientific contribution of our work is that our method of correlating task-oriented morphology and haptic-modulated control i) allows for distributed sensing and interaction along the robot's cage – instead of the tip of the end-effector as in previous works – and ii) simplifies the formalization of the controller itself, as it eliminates the need for high-level logics, re-tuning, contact model and constraints, or prior knowledge of the environment. Both aspects have limited previous works to structured environments and localized interaction at the end-effector, discarding complex environments that exhibit an elastic behavior and can touch the drone anywhere around its body. We have extended the “Introduction” chapter accordingly.

Revision:

P. 1, Line 39:

Despite considerable ~~progress in~~ **advancements over** the last decade [7], the design and control of aerial robots still face limitations when interacting with compliant obstacles. ~~First, Common approaches for aerial physical interaction (APhI), such as impedance and admittance controllers, are easy to implement but are mainly tailored for exerting desired forces on rigid surfaces [R1, R2, R3]. In contrast, traversing vegetation necessitates the use of different modes of interaction~~ **interaction modes**, including pushing to bend obstacles, sliding along them, or employing a combination of these techniques. ~~Current drones~~

~~transition between different interaction modes through high-level switching policies.~~ Model-based or robust controllers enable different modes of interactions as demonstrated in complex tasks such as making contact and pushing hinged doors or rolling carts [R4, 11, 12], and pushing and sliding along surfaces for writing or inspection [R5, 14, 15]. However, ~~such switching policies are commonly these solutions require high-level switching policies~~ based on empirically tuned conditions and thresholds, which become complicated to define when mechanical properties of the environment are stochastic and complex to model, as is the case with vegetation. Moreover, while ~~physical interaction with static~~ extensive research has been dedicated to APhI with rigid and movable obstacles ~~has been extensively studied~~ [17], obstacles with an elastic response, such as vegetation, have received limited attention. Another limitation of current drones is that interaction tasks are constrained to sensorized end-effectors, as demonstrated by tasks such as sensor installation and retrieval [18, 19], contact-based inspection [20, 21], or object manipulation [22, 23]. However, traversing vegetation demands an “unconstrained” interaction, where ~~in~~ contacts can be established and moved along the entire body of the drone ~~as it traverses the vegetation during traversal~~. As a result, haptic sensing cannot be confined to the end-effector but must ~~instead~~ be distributed along the entire body of the robot.

P. 2, Line 70:

Translating these insights in task-oriented morphology and minimalistic controllers, researchers are developing ground robots that use their sensorized body to skillfully push aside and slide through ~~cluttered obstacles of a large range of stiffness~~ compliant obstacles [26, 27, 32], ~~and~~ drones that exploit ~~circular~~ protective cages to fly through rigid obstacles ~~without exteroceptive sensing~~ [33, 34], and foldable drones that traverse narrow passageways [R6, R7].

New references added:

R1: Augugliaro, F. & D’Andrea, R. Admittance control for physical human-quadrocopter interaction. 2013 European Control Conference (ECC) 1805–1810 (2013).

R2: F. Ruggiero, J. Cacace, H. Sadeghian and V. Lippiello, "Impedance control of VTOL UAVs with a momentum-based external generalized forces estimator," 2014 IEEE International Conference on Robotics and Automation (ICRA), Hong Kong, China, 2014, pp. 2093-2099, doi: 10.1109/ICRA.2014.6907146.

R2: Yüksel B, Secchi C, Bühlhoff HH, Franchi A. Aerial physical interaction via IDA-PBC. The International Journal of Robotics Research. 2019;38(4):403-421. doi:10.1177/0278364919835605

R4: Lee, D., Seo, H., Kim, D., & Kim, H. J. (2020, May). Aerial manipulation using model predictive control for opening a hinged door. In 2020 IEEE International Conference on Robotics and Automation (ICRA) (pp. 1237-1242). IEEE.

R5: Khedekar, N., Mascarich, F., Papachristos, C., Dang, T. & Alexis, K. Contact-based navigation path planning for aerial robots. 2019 International Conference on Robotics and Automation (ICRA) 4161–4167 (2019).

R6: Patnaik, K., Mishra, S., Sorkhabadi, S. M. R. & Zhang, W. Design and control of squeeze: A561 spring-augmented quadrotor for interactions with the environment to squeeze-and-fly 1364–1370562 (2020).

R7: Fabris, A., Aucone, E. & Mintchev, S. Crash 2 squash: An autonomous drone for the traversal of 564 narrow passageways. Advanced Intelligent Systems 4, 2200113 (2022).

2. NMPC design:

Thank you so much for your questions. We address each of them below.

- how the reference force and torque are determined in (5)?

Response:

Equation 5 is written in a general form, where reference values for thrust and torque can be included; in our scenario, we set these values equal to the hovering thrust and to null torques, so that the drone keeps a near hovering condition during both free-flight and physical interaction – this is in accordance to the quasi-static motion that we want to guarantee.

Revision:

P. 10, Line 447:

... finally angular velocity, thrust, and torque terms act as regularization term (reference thrust set to hovering value, and angular velocity and torques set to zero, valid for both free-flight and interaction).

- How is the horizon N determined in the NMPC design, especially given the unknown obstacles? It will also be important to perform some sort of ablation study on the impact of NMPC parameters to the performance of the overall system.

Response:

The simulation environment that we developed (based on ROS and Gazebo 3D simulator) was very useful to properly tune the prediction horizon and the controller parameters, before transferring the strategy on the real drone. We clarify below the choice of the prediction horizon and conduct an additional ablation study to assess the impact of the controller parameters on the performance of the system.

The prediction horizon was determined over multiple simulations: we tuned the horizon length N (steps) by gradually increasing it, finding a value high enough to allow to always find a feasible solution – larger prediction horizons help to guarantee stability and feasibility of the optimization problem – and small enough not to increase the computation time. Moreover, we used a quantitative metric based on the average distance between predicted trajectory along the horizon and the quadcopter's real trajectory. This metric provides the goodness of both the NMPC implementation (to validate that the model and the constraints are feasible and properly defined) and the solver algorithm.

To quantitatively assess the impact of the controller parameters on the performance of the NMPC in terms of trajectory following and safe physical interaction (which are the two main objective of the controller), we performed an ablation study in simulation for different weights included in the optimization cost vectors. We considered 5 metrics related to the tracking of the lateral position and the forward velocity, as well as the stability of the drone during the interaction in terms of applied force and attitude oscillations – the same metrics introduced in the Results and Methods chapters. We defined 3 sets of weight: i) weights related to the path following higher (at least 2 order of magnitude) with respect to the weights related to the interaction (impedance), ii) balanced weights (same order of magnitude or 1 order different), iii) weights related to the path following lower (at least 2 order of magnitude) with respect to the weights related to the interaction (impedance). We performed 5 experiments for each set, and for each value of stiffness (low, mid, high). A detailed discussion of this analysis and the obtained results is presented in a new section of the Supplementary Information called "Impact of NMPC Objective Cost Weights".

Revisions:

P. 24, Line 674:

We developed and validated the NMPC and the overall strategy using the Robot Operating System (ROS) and Gazebo 3D simulator (Supplementary Fig. 2A). We used the RotorS [38-1] package for simulating a quadrotor and its flight architecture, to which we added an F/T sensor plugin and the CAD of the cage – the visual model of the cage is a sphere but the interaction occurs only on the ring and the inertial parameters corresponds to the real-world prototype (Supplementary Tab. 1). We used a Gazebo plugin to generate an elastic behavior of the environment, by using one rotational spring joint. We performed several experiments in the 3D physical simulator for implementation, debug, and testing tuning and evaluation.

P. 24, Line 700:

~~Discussion about~~ Stability and Feasibility of the NMPC-based Controller

The stability and feasibility of receding horizon problems are generally not guaranteed except in cases of infinite horizon control. If the prediction horizon is limited to N steps, stability and feasibility are questionable. Thus, we opted for a finite receding horizon for our approach. In principle, it has been shown that under the assumption that the reference trajectory is consistent with the vehicle dynamics, the stability and feasibility is guaranteed by selecting a sufficiently large prediction horizon length [40-3], at the cost of a higher computation effort. Further, terminal cost and terminal constraints can be chosen such that closed-loop stability and feasibility are ensured [41-4]. For applications on aerial robot, the computation power is often limited by the platform, yet fast control action are needed, so it is important to find a proper balance between the two aspects.

For our work, we decided for use a terminal cost, as it is easy to compute and its weights can be directly tuned, whereas terminal constraints are generally more difficult to solve [42-5]. The approach has been validated in several work, e.g. for path following in [43-6], where the authors provided closed-loop asymptotic stability of MPC without stabilizing constraints, as well as for physical interaction with a UAV [44-7]. ~~We heuristically performed several simulations with different values of horizon length and weights, achieving a trade-off between stability performance and computational load. We tuned such parameters and evaluated the performance for different values of stiffness in the environment. It is worth noticing that stability is further simplified since our strategy does not use switching conditions between free flight and contact, as well as between the interaction modes. Finally, since our~~ Since our problem relates to path following, the reference trajectory is defined by the forward velocity and the position in the other two axes, which is feasible with respect to the limits and the dynamics of the system. By combining a low reference velocity (quasi-static motion) and the constraints on the orientation (which act as an additional safety margin), we ensure that the drone did not have aggressive behavior and the controller could find a feasible solution – i.e. the NMPC has no hard constraints, only limits on the angles and thrust, so the OCP is even easier to solve, as it just derives how the external wrench affects the dynamics and what is the desired impedance acceleration in accordance to the external stimuli, while keeping track of following the path in parallel. To quantitatively evaluate the stability and feasibility, we performed several simulations with different values of horizon length, achieving a trade-off between stability performance and computational load. In detail, we tuned the horizon length N (steps) by gradually increasing it, finding a value high enough to allow to always find a feasible and stable solution and small enough not to increase the computation time. In parallel, we monitored in simulation the distance between the predicted trajectory over the horizon and the executed trajectory (Supplementary Figure 2B), as well as the value of the cost function, to be sure that the NMPC could effectively find an admissible solution at all the iterations. This metric helps to check for eventual problems in both the NMPC implementation (to validate that the model and the constraints are feasible and properly defined) and the solver algorithm. Finally, it is worth noticing that the feasibility and stability of the NMPC solution in our approach are further simplified since our strategy does not use switching conditions between free-flight and contact, as well as between the interaction modes –

switching conditions in the optimization problem may create discontinuities in the solution, which are generally mitigated with high-level logics to allow smooth transitions [8].

P. 30:

Supplementary Figure 2: **Simulation tools exploited to validate the stability and feasibility of the controller, and to evaluate its performance in terms of optimization. A** Gazebo 3D environment developed to test our solution. **B** Plot of the distance between predicted and real, executed trajectory over time, from an experiment where the parameters were properly tuned.

P. 26, Line 773:

Impact of NMPC Objective Cost Weights

To quantitatively assess the impact of the controller parameters on the performance of the system in terms of path following and safe physical interaction – the two main objectives of the NMPC – we perform an ablation study in simulation for different weight parameters included in the optimization cost vectors (Supplementary Fig. S3 and Supplementary Movie S5).

We use 5 metrics related to the tracking of the lateral position and the forward velocity, as well as the stability of the drone during the interaction, in terms of applied force and attitude oscillations (on both roll and pitch) – see section Statistical Analysis for details about the metrics. We defined 3 sets of parameters: i) **High Traj, Low Imp** – weights related to the path following higher (at least 2 orders of magnitude) with respect to the weights related to the interaction (impedance), ii) **Mid Traj, Mid Imp** – balanced weights (same order of magnitude or 1 order different), iii) **Low Traj, High Imp** – weights related to the path following lower (at least 2 orders of magnitude) with respect to the weights related to the interaction. We performed several experiments for each set and for each value of stiffness (*low, mid, high*), to calculate the success rate, and conducted a statistical analysis on a subset of 5 experiments to assess the impact of the weight parameters on the performance of the system.

As depicted in the Supplementary Fig. 3, when the path following has a higher weight in the optimization (**High Traj, Low Imp**), the errors in the lateral position are smaller (Suppl. Fig. 3A). However, during contact the drone interacts more aggressively, due to the penalization of the impedance term, and sometimes cannot safely dampen the elastic behavior of the environment. The unwanted behavior during the interaction can be seen by higher values of longitudinal velocity error (Suppl. Fig. 3B), higher longitudinal force (Suppl. Fig. 3C), as well as higher oscillations on the attitude (Suppl. Fig. 3D and E). This effect increases when the obstacle gets stiffer, as it has a stronger response upon interaction. In this scenario, the success rate was 100%, 80% and 40% for *low, mid, and high* stiffness respectively (Suppl. Fig. 3F).

Viceversa, when the impedance weights have a higher impact with respect to the path following ones (**Low Traj, High Imp**), the stability greatly improves (Suppl. Fig. 3C, D, E) but the performances in tracking the reference path degrade; the errors in the lateral position, for each stiffness, are the highest among the three sets, and almost reach 0.5m in magnitude (Suppl. Fig. 3A). Success rate was 100% for all the stiffness values. Balancing the weights for the two objectives (**Mid Traj, Mid Imp**) still allows

to ensure a safe behavior during the interaction (Suppl. Fig. 3C and D) and to reduce the errors in tracking the reference path (with respect to the previous set of weights – Suppl. Fig. 3A and B), finding a good compromise between the two objectives.

P 31:

Supplementary Figure 3: Ablation study to evaluate the impact of the cost weights on the controller performances. **A** Tracking error of the lateral position; **B** longitudinal velocity; **C** max longitudinal force; **D** oscillations on the roll motion; **E** oscillations on the pitch motion. Mann-Whitney U test performed among the three sets and stiffnesses; box plots indicate median (middle line), 25th, 75th percentile (box) and 5th and 95th percentile (whiskers) as well as outliers (single points); number of tests used for the statistical analysis = 5 for each stiffness and set of weights; $**P < 0.01$ for all the boxplots compared among the sets, for each stiffness - asterisks not added to the plot for clarity of the figure. **F** Success Rate (%) for the different values of stiffness and different set of weights.

References used in the text above (previously part of the References, now part of the Supplementary Bibliography as indicated in the guidelines):

[1] Furrer, F., Burri, M., Achtelik, M. & Siegwart, R. Rotors – a modular gazebo mav simulator framework. *Studies in Computational Intelligence* 625, 595–625 (2016).\\

[3] Grüne, L., Pannek, J., Seehafer, M. & Worthmann, K. Analysis of unconstrained nonlinear mpc schemes with time varying control horizon. *SIAM Journal on Control and Optimization* 48, 4938–4962 (2010).

[4] Borrelli, F., Bemporad, A. & Morari, M. *Predictive Control for Linear and Hybrid Systems* (Cambridge University Press, Cambridge, 2017).

[5] Mayne, D., Rawlings, J., Rao, C. & Scokaert, P. Constrained model predictive control: Stability and optimality. *Automatica* \36, 789–814 (2000).

[6] Mehrez, M. W., Worthmann, K., Mann, G. K. I., Gosine, R. G. & Faulwasser, T. Predictive path following of mobile robots without terminal stabilizing constraints. *IFAC-PapersOnLine* \50, 9852–9857 (2017).

[7] Kocer, B. B., Tjahjowidodo, T. & Seet, G. G. L. Model predictive uav-tool interaction control enhanced by external forces. *Mechatronics* 58, 47–57 (2019).

[8] L. Peric, M. Brunner, K. Bodie, M. Tognon and R. Siegwart, "Direct Force and Pose NMPC with Multiple Interaction Modes for Aerial Push-and-Slide Operations," 2021 IEEE International Conference on Robotics and Automation (ICRA), Xi'an, China, 2021, pp. 131-137, doi: 10.1109/ICRA48506.2021.9561990.

- How sensitive is the controller to the accuracy of wrench measurement and system modeling? Lastly, as the authors mentioned in the paper, when the drone starts and exits physical contact with the environment, there is a jump in the external force which may cause instability. How does the NMPC handle this stability issue?

Response:

Regarding the system modeling, in our approach the only parameters required for the dynamic model of the system used in the optimization are the inertial properties of the drone (mass and moments of inertia), which are measured and computed from our CAD respectively. Since the drone operates in quasi-static motion and the main contribution in the rotational dynamics (last row of Eq. 1) comes from the external torques, uncertainties on the moments of inertia have a small impact during the interaction. Accordingly, inaccuracy in the mass just results in a difference of the thrust force that the drone must generate.

Regarding the external wrench, differently from current approaches in the field of aerial physical interaction, the task of traversal does not aim to track force and position accurately by careful planning of movements, but rather to safely handle interaction with environments with an elastic response. Our approach relies on the external wrench in an impedance-fashion to dampen the oscillations of the environment, thanks to the specific objective term. In such a way, the controller directly dampens the response of the environment by exploiting every increase or decrease in the wrench. However, as correctly highlighted by the reviewer, in case of wrench measures totally offset from the real quantities involved during the interaction (e.g., force/torque sensor not calibrated), the drone might have a response that is quite weaker or stronger than it should: in the first case, the drone would push less, but then this action would generate an error in the velocity following, with a consequent increase in the push; in the second case, instead, the drone might push too much, potentially leading to instability during the interaction with the environment. Therefore, any inaccuracy or biases must be removed. For the reasons explained above, as well as to generally avoid jumps at the start and end of the interaction, we add a filter for the force/torque sensor reading in the loop. In this way, during the whole interaction the signal that the NMPC receives is smoother, without abrupt changes or bouncing conditions especially upon the establishment of a contact and the detachment (see Fig. 4). In addition, the filter also removes biases by using the measures during the hovering condition. It is important to mention that the filter, however, needs to be complemented with the quasi-static motion.

We apologize for not clarifying these points already at the first submission, we have now added them in the sections “Drone Architecture” and “Runtime Discussion”, both part of the Suppl. Information.

Revisions:

P. 24, Line 652:

... has a total mass of 1.2 kg. Inertial parameters useful for the dynamical model inside the NMPC are measured (mass) or obtained from the CAD (moments of inertia), and reported in Supplementary Tab. 1.

P. 33:

Inertial Parameters	Values
Mass [kg]	1.200
Inertia along x-axis [kg m ²]	0.007
Inertia along y-axis [kg m ²]	0.007
Inertia along z-axis [kg m ²]	0.012

Supplementary Table 1: Inertial parameters of the designed aerial robot.

P. 25, Line 693:

At runtime, for each iteration, the NMPC receives the data from the sensors (i.e. the external wrench and the state), sets the initial state condition using the current values of the state, and integrates the continuous dynamics in order to discretize it; ~~then, the~~. In our approach, the only physical parameters required for the dynamics of the system used in the optimization are the inertial properties of the drone (mass and moments of inertia, Supplementary Table 1). Since the drone operates in quasi-static motion and the main contribution in the rotational dynamics (last row of Eq. 1) comes from the external torques, variations on the moments of inertia have a small impact during the interaction. Accordingly, inaccuracy in the mass just results in a difference of the thrust force that the drone has to generate. Then, the algorithm gets the upper and lower bounds for the constraints, as well as the objective weights, and computes the cost vectors. ~~;~~ Thus, it starts the optimization from a virtual hovering condition (at the first initialization) or from the previous NMPC solution (for the following iterations); once the optimization converges, the NMPC returns both control \mathbf{u} and state variables \mathbf{x} along the whole prediction horizon.

P. 25, Line 717:

As common practice, we choose to keep the wrench constant (zero derivative) during the prediction (1 second). Nevertheless, the NMPC internally updates the external wrench with a new value every 0.05 seconds and performs the optimization again. Without loss of generality, this assumption is thus justifiable i) because the reference velocity we target is really small (-0.15 m/s), making the quadrotor motion quasi-static, and ii) because the average variation between two updates of external wrench is quite limited throughout the execution of the task (variation of longitudinal force within 0.05 seconds is on average around 0.02 N). To further avoid jumps in the wrench measures, and consequent bouncing effects, we add a filter in the loop to smoothen the external wrench received by the NMPC. This guarantees to avoid abrupt increases and decreases of the wrench, which is beneficial especially during first contact and detachment (i.e., when moving from a zero to non-zero wrench). The filter further removes eventual biases by exploiting measures during the hovering condition. The usage of such approach makes the overall system less sensitive to inaccuracies in the direct wrench measurements. It is worth to mention that this might be a limitation in scenarios with a rapidly varying external wrench or when the quasi-static motion is not guaranteed, potentially affecting the stability of the system or the performance of the interaction.

3. The proposed controller relies on the integrated force/torque sensor to understand the interaction with the environment – this means the contact must take place in the designated area where sensor measurement can be taken. This may not be the case for traversing through forests with branches and leaves. Some discussions should be included to discuss this limitation.

Response:

Haptic sensing is realized by connecting the cylindrical shell to the robot's frame using a six-axis load cell. This design facilitates the measurement of the net wrench resulting from contacts across the entire surface of the shell, effectively addressing the 2D interaction studied in this article. Introducing a spherical protective structure, in place of the cylindrical shell, would enable extended haptic sensing all around the robot. This enhancement is crucial for providing the haptic information required to navigate environments with a complex 3D distribution of obstacles, such as forests.

We extended the “Discussion” chapter to clarify these aspects as suggested by the reviewer.

Revision:

P. 7, Line 363:

In the current prototype, the discoid shell leaves the propellers vulnerable to collisions with branches, twigs, and leaves from both above and below the drone. To mitigate this, a spherical protective cage can be employed. For instance, the authors have successfully utilized a hemispherical cage to safeguard the drone's propellers during branch landings [3].

...

The existing haptic sensing method measures the net wrench acting on the drone, which fails to discern the locations of individual contacts on the body and accurately measure local interaction forces. Preliminary simulation results indicate that this limits the robot's ability to navigate environments with multiple compliant obstacles (refer to Supplementary Information and Supplementary Movie S6). In scenarios where the drone attempts to traverse both soft and rigid obstacles simultaneously, it may become stuck on rigid obstacles due to its inability to detect paths of lower stiffness with a higher likelihood of traversability. Addressing this challenge requires higher-resolution haptic sensing, achievable through electronic skins [41], visual haptic sensors or whisker arrays [42], to estimate the compliance and traversability of obstacles. When combined with high-level path planning, drones could use this information to plan trajectories towards more traversable areas. This integration, together with the extension of the haptic controller from 2D to 3D, will be crucial to effectively manage multiple interactions and to deploy the drone in complex natural environments.

4. I notice that there is a cable connected to the drone in all experimental videos. Is that cable for data communication or power? It introduces some concerns that the cable may either obstruct the motion of the drone or provide stability support to the drone under disturbance. The authors need to ensure that there is no force at any time during the experiment.

Response:

Thank you so much for pointing it out. The cable attached to the drone is a safety rope in case the drone drops or must be stopped. The cable is slack during the experiments, as shown in the videos, so it does not provide support or disturbance during flight.

5. How do you determine the three different values of stiffness in experiments? This may have major impacts on the results shown in Fig 5 and Fig 6 so this point should be made clear.

Response:

Our plate setup includes a foam plate connected to a vertical aluminum profile with two hinges, which allows the plate to rotate around one axis (vertical one). To give an elastic behavior to such rotational joint, we attached linear springs at the hinges. Since the two hinges are in parallel the contribution in

terms of stiffness is the sum of the two. By fixing the radius of the hinge we changed the resultant torsional stiffness of the joint by changing the linear springs. We extended the “Compliant Hinged Plate” section with additional explanation, a figure, and a table.

Revision:

P. 24, Line 656:

The hinged surface used during the experimental validation is a lightweight construction foam plate, with a weight of around 400 g, and dimensions of 60 cm by 80 cm by 2 cm. ~~A spring in the hinge~~The plate is attached to a fixed structure via two hinges, which allow the foam plate to rotate around one axis (vertical one), forming a single rotational joint (Supplementary Fig. 1A). Each of the hinges has a rotating part connected to a linear spring with stiffness k_{spring} that gives the obstacle a compliant behavior, as it returns the surface to its initial position and provides the force which the quadrotor has to counteract (Supplementary Fig. 1B). In detail, since the springs are in parallel, the linear stiffness of the rotational joint is the sum of the two contributions $k_{linear} = k_{spring1} + k_{spring2}$. By fixing the radius of the hinge (in our case $R_{hinge} = 6\text{ mm}$), the equivalent torsional stiffness $k_{torsional}$ of such rotational joint (Supplementary Fig. 1C) can be changed by changing the linear stiffness k_{linear} (thus the springs themselves), according to the following relation: $k_{torsional} = k_{linear} * R_{hinge}^2$. For our experiments, the effective torsional stiffness values range across 1 order of magnitude (Supplementary Tab. 2), with initial point of contact located about 60 cm from the axis of the hinge.

P 29:

Supplementary Figure 1: **Hinged surface used during the experiments.** **A** Foam plate attached to a rigid pole via two hinges. **B** Single hinge in the resting condition and when the plate is deflected. **C** Diagram of a single hinge mechanism.

P 34:

k_{linear} [N/mm]	$k_{torsional}$ [N mm/rad]
0.5	18
2.16	77.76
4.32	155.52

Supplementary Table 2: **Linear and torsional stiffness of the rotating joint during the experiments.**

6. For the non-streamlined design in the experiments, I understand that it does not work because it cannot slide. However, it seems that it can just push the obstacle away if the drone follows the original path (instead of maintaining contact to the obstacle). I guess it is just because this design does not fit the controller design, so the current conclusion can be a bit misleading.

Response:

The traversal with the square cage fails because the drone does not transition from the Pushing to the Sliding mode. This failure is attributed to the shape of the cage, causing the drone to yaw towards the hinged surface, pushing it and following its rotation. To prevent the drone from getting stuck in the Pushing mode, we would need to define thresholds to enforce the transition into the Sliding mode. However, such an approach conflicts with our strategy, as our work aims to demonstrate that successful traversal can be achieved without high-level switching policies but, instead, with a task-oriented morphology. We discuss these aspects in the revised text.

We clarified it in the “Ablation Study” section accordingly.

Revision:

P. 6, Line 270:

~~First, we adopted~~Initially, we adopt a squared cage and conducted experiments for each of the three values of stiffness. ~~Due to the~~The non-streamlined ~~nature of the body shape, the drone gets constrained in orientation along the vertical axis, upon contact, and~~ shape of the cage leads the yaw rotation of the drone to become constrained with the rotation of the surface. Consequently, the drone remains stuck in the Pushing mode ~~consequently~~ and could not slide or realign with the straight path (test for *mid* stiffness in Fig. 6A). It is worth noting that defining thresholds, as done in previous works, might facilitate the transition into a lateral Sliding mode. However, this approach contradicts our strategy, which aims to demonstrate that successful traversal can be achieved using optimized body morphologies without the need for empirically tuned thresholds and switching conditions.

7. In Figure 6, the figure has the haptic feedback off for C. I am confused how the controller will work without the force/torque sensor measurement. Some further explanations may be helpful.

Response:

Thank you for your comment. With that set of experiments, we indeed wanted to show the importance of haptic feedback in the loop, which is an essential component for robot performing physical interaction. With feedback off, the controller enables the drone to follow the reference straight trajectory, but the impedance term in the controller is not active due to the absence of haptic feedback. The role of such a term is to appropriately suppress oscillations caused by the elastic response of the obstacle without the need for a model. For low stiffness obstacles, trajectory tracking, coupled with the low friction surface and streamlined shape of the shell, is sufficient for successful traversal. However, as the stiffness of the obstacle increases, the lack of haptic-driven control causes the drone to fail due to the stronger reaction of the environment during interaction. In this condition, the “haptically blinded” drone fails – ultimately loses stability – because it can neither use the feedback to push the obstacle nor dampen the oscillations caused by stiff obstacles.

We have clarified this point in the “Ablation Study”.

Revision:

P. 6, Line 286:

~~Last, we conducted~~Lastly, we conduct experiments to demonstrate the importance of ~~sensing the external wrench~~ haptic sensing to inform the feedback control loop, ~~for both monitoring~~ both to monitor the effects of the wrench on the ~~dynamics of the drone~~ drone’s dynamics and for reacting to the interaction with the environment during traversal. ~~In detail, we performed experiments neglecting the external wrench~~ To this end, we disable the haptic sensing, thereby removing the wrench measurements from the dynamics model and ~~removing the desired~~ the impedance term in the NMPC formulation, ~~i.e. treating the interaction~~. This allows the controller to track the reference trajectory, but the interaction is

~~treated only~~ as a disturbance (test for mid stiffness in 6C). For *low* stiffness ~~the task was accomplished successfully~~ obstacles, trajectory tracking, coupled with the low friction surface and streamlined shape of the shell, is sufficient for successful traversal due to the ~~light~~ weak mechanical response of the environment, ~~yet for~~ (small wrenches involved and negligible oscillations). However, as the stiffness of the obstacle increases, the lack of haptic-driven control causes the drone to fail. For *mid* and *high* ~~the drone failed~~ stiffness, the drone fails to traverse the obstacle (reaching instability) in 70% and 100% of the experiments respectively (success rate in Fig. 6D), ~~due to the higher wrenches and notable oscillations~~. These experiments ~~show how the robot cannot rely only~~ demonstrate that the drone cannot ~~rely solely~~ on morphology-enabled behaviours to successfully traverse stiffer obstacles, ~~and confirm the~~. They confirm the crucial role of the ~~controller in opportunistically dampen~~ closed-loop haptic feedback controller in sensing and dampening the oscillations induced by the ~~obstacle-compliant~~ obstacles during the interaction.

8. The result shown in Figure 7 is with only one branch of leaves – what will happen if there are multiple contacts simultaneously? It seems that the force/torque sensor will provide a net wrench and it might confuse the controller. Some additional discussions and experimental results will be needed to make this point clear.

Response:

In the Discussion included in our original article, we already emphasized the necessity for a high-resolution haptic sensing system capable of measuring the interaction force of each individual contact, rather than focusing solely on the net wrench caused by these contacts on the drone. To verify this assumption, we conducted a series of experiments with our simulator. Even with just two obstacles, it became evident that certain configurations allow traversal, while others result in the drone becoming stuck, particularly when facing stiffer obstacles. These findings highlight the need for future investigation to equip the drone with decision-making capabilities, mostly based on enhancing the perception system and developing planning strategies to replan the trajectory based on the compliance properties of the obstacles (e.g. planning trajectories towards more compliant areas).

We have revised the “Discussion” chapter to better highlight the challenges associated with multi-contact sensing. Additionally, a dedicated section has been included in the Supplementary Information, focusing specifically on the traversal of multiple obstacles.

Revision:

P. 7, Line 369:

The existing haptic sensing method measures the net wrench acting on the drone, which fails to discern the locations of individual contacts on the body and accurately measure local interaction forces. Preliminary simulation results indicate that this limits the robot's ability to navigate environments with multiple compliant obstacles (refer to Supplementary Information and Supplementary Movie S6). In scenarios where the drone attempts to traverse both soft and rigid obstacles simultaneously, it may become stuck on rigid obstacles due to its inability to detect paths of lower stiffness with a higher likelihood of traversability. Addressing this challenge requires higher-resolution haptic sensing, achievable through electronic skins [41], visual haptic sensors or whisker arrays [42], to estimate the compliance and traversability of obstacles. When combined with high-level path planning, drones could use this information to plan trajectories towards more traversable areas. This integration, together with the extension of the haptic controller from 2D to 3D, will be crucial to effectively manage multiple interactions and to deploy the drone in complex natural environments.

P. 27, Line 807:

Multiple Obstacles Traversal

We have performed experiments in our simulation environment to investigate the behavior of the system during interaction with multiple obstacles (Supplementary Fig. 4 and Supplementary Movie S6). This is an initial investigation to draw insights into future developments.

During contact with multiple obstacles, the force/torque sensor only measures the net wrench acting on the drone. Due to the implementation of the strategy, the controller is consequently informed with a single wrench. We have experimentally tested that the drone might be able to traverse the obstruction in some cases, or fail, depending on the stiffness and location of the obstacles with respect to the drone itself and to each other.

First, we investigated a scenario where the drone (and the path to follow) is centered with respect to two obstacles, aligned and spaced apart from each other (Supplementary Fig. 4A) – we tested for obstacles having either same or different stiffness. In this scenario, the drone enters in contact with both obstacles simultaneously and the net wrench resulting from the two contacts is informative enough to guide the drone towards the other side of the obstruction: for obstacles with equal stiffness, the drone simply proceeds straight along the path as the contribution of the lateral forces cancels out; during interaction with obstacles having different stiffness, instead, the lateral component of the resultant wrench will guide the drone towards the most compliant obstacle, as if the interaction would be with a single obstacle.

A second scenario involves two obstacles misaligned with respect to each other (Supplementary Fig. 4B). We noticed that the drone starts to interact with the first obstacles, but upon contact with the second one failure cases might happen for some misalignments and if the second obstacle is very stiff (almost rigid): indeed, once the drone pushes aside the first obstacle and enters in contact with the second, rigid one, it might get stuck due to the strong contribution of the second wrench on the net wrench, which is opposite to the path direction. This general scenario is likely to be encountered in natural environments, as obstacles have different compliance and are generally spaced in a very heterogeneous way. Thus, such insights highlights that future investigation would require to include decision-making capabilities onboard, mostly based on i) enhancing the perception system to detect and locate multiple contacts, as well as estimating the compliance of different obstacles, and ii) developing planning strategies that exploit such information to adapt the path online – e.g. using the information on the location of contacts and map the compliance of the environment to move towards more compliant areas.

P. 32:

Supplementary Figure 4: **Traversal of multiple obstacles.** **A** Example of a successful scenario where two obstacles are aligned and equally spaced apart from the reference path. **B** Example of a failure case due to the rigidity of one of the two obstacles and the misalignment with respect to each other.

Reviewer #3

This paper presents an approach for aerial robots to traverse unknown compliant obstacles using direct physical interaction. The combination of the task-oriented design and haptic feedback control shows promise for improving aerial robots' ability to move through dense vegetation.

Response.

We thank the reviewer for the positive evaluation of our manuscript. Our answers and revisions are reported below.

In the following we report the main revisions in the manuscript. The direct answers to the reviewers' comments are presented in blue font, whereas the newly added content to the Main article is presented in red font and the removed content is also ~~strikethrough~~.

I have a few main comments:

- The introduction motivates the problem well, but is a bit lengthy. I would suggest tightening up the introduction to more clearly highlight the key gaps this work aims to address.
- The writing could be tightened up in some areas - there is occasional verbose language that makes the paper longer than necessary.

Response:

Thank you for the valuable feedback, we have removed unnecessary repetitions and verbose language throughout the article, especially in the Introduction, the Results, and the Discussion chapters, which have been modified also in accordance with the comment other reviewers. The changes are directly reported in the revised manuscript.

- In the results, it would be good to provide some quantitative metrics on the tracking performance for different stiffness values, rather than just the qualitative analysis. This would better showcase the versatility of the approach across varying compliance.

Response:

The original analysis of the performance summarized in Figure 5 was already quantitative, providing a statistical analysis based on nine traversal tests for each stiffness value (low, mid, and high). The analysis considered three metrics: lateral position error, longitudinal velocity, and max longitudinal force. However, recognizing that we may not have clearly conveyed this message in the text, we have extended and revised the "Experimental Validation" section (Results) and the "Statistical Analysis" section (Methods). In the revised sections, we provide a clearer explanation of how the quantitative metrics used for the comparisons are computed. Additionally, we introduce two additional metrics for assessing the stability of the system, particularly focusing on the oscillations occurring in the attitude (both roll and pitch) during the interaction with the compliant environment.

Revision:

P. 5, Line 222:

~~Further, we~~We demonstrate the versatility and repeatability of the proposed strategy by performing 10 experiments for each of the three stiffness values (Fig. 5 and Supplementary Movie S2). During all the experiments, the reference speed ~~was is~~ set to 0.15 m/s. The ~~following~~ quantitative analysis in Fig. 5 refers only to the physical interaction, i.e., between the first contact and the detachment. ~~The tracking of the lateral position p_y is reported in Fig. 5A, which shows that the error (expressed as absolute~~

tracking error $e_y = \sqrt{(\sum_{k=1}^M (p_{y,k} - p_{y,ref})) / M}$ with M equal to the number of samples during the interaction) is kept relatively close to 0, and the medians are comparable. The success rate (Fig. 5F) shows the number of successful tests out of ten. Since there is one failure for the highest stiffness, the rest of the metrics are evaluated on nine experiments to have the same number of data points for each stiffness. The five quantitative metrics for such analysis are explained in detail in the Methods chapter. The performance in terms of tracking of reference path is assessed by computing errors in lateral position and longitudinal velocity. Statistical analysis using the Mann-Whitney U test does not indicate a significant difference across different stiffness values (P value > 0.5). As depicted in Fig. 5A, instead, refers to the tracking of the longitudinal velocity v_x (aligned with the reference path), which is close to the reference value of 0.15 m/s (dashed line) for the different values of stiffness. For both position errors and velocity tracking, the statistical analysis does not show a statistical difference between the distribution of data (Mann-Whitney U test, P value > 0.05), the drone does not experience drifts exceeding 0.2m during the interaction. Although the data exhibit greater dispersion as stiffness increases, the median values remain similar for the three stiffness levels, specifically at 0.084 m, 0.098 m, and 0.090 m, respectively. Figure 5B shows non-zero velocity errors, which are expected as the drone adjusts its speed during interaction with the environment. The medians of the velocity errors are similar across distributions, measuring 0.108 m/s, 0.104 m/s, and 0.109 m/s, respectively. The maximum force values along the longitudinal axis $\max(F_{ext,x})$ are depicted in Fig. 5C. Values are higher for increasing stiffness, since the drone has to increase with stiffness, reflecting the drone's need to push more to traverse stiffer obstacles, as expected. The statistical analysis indicates. The median of the force is equal to 1.193 N, 1.272 N, and 1.670 N for the three values of stiffness. Statistical analysis reveals that the distributions of data, only specifically for the low and high stiffness values, are statistically different between each other, from each other with a significance level of 95% (Mann-Whitney U test, P value < 0.05). The stability of the drone during interaction is assessed by analyzing the attitude oscillations (Fig. 5D and E). The amplitude of oscillations is calculated using the root mean square (RMS) of the roll angle ϕ and pitch angle θ . Across the three values of stiffness, the drone exhibits minimal oscillations, with medians well within the admissible roll and pitch angle limits [-20, 20] deg. Specifically, the medians are reported as 2.664 deg, 2.100 deg, and 3.525 deg for the roll and 3.935 deg, 5.057 deg, and 5.828 deg for the pitch, respectively. Statistical analysis for roll oscillations indicates no significant difference among the data distributions (Mann-Whitney U test, P value > 0.05). In contrast, the distributions of data for pitch oscillations show statistical difference for some stiffness values (Mann-Whitney U test, * P < 0.05 between *low* and *mid*, and ** P < 0.01 for *low* and *high*). This observation aligns with the expectations, as the drone needs to pitch more to generate additional force when encountering obstacles with higher stiffness. The presented results imply that the performance of our approach in traversing compliant obstacles is consistent and independent of the obstacle compliance in the tested range, as further verified by the fact that the drone was capable of traversing in almost all the experiments (success rates over 10 experiments reported in 5D-F).

Figure 5: **Performance of the NMPC in tracking the reference path, and external force experienced terms of path following and stability during the interaction phase for different values of stiffness.** **A)** tracking error of the lateral position; **B)** longitudinal velocity; **C)** max longitudinal force; **D)** oscillations on the roll attitude; **E)** oscillations on the pitch attitude. Mann-Whitney U test performed for all possible combination of distribution; **median (red box plots indicate median (middle line), 25th, and 75th percentiles reported; Number-percentile (box) and 5th and 95th percentile (whiskers) as well as outliers (single points); number of tests used for the statistical analysis = 9 for each stiffness (over a total of 10 experiments each); $P > 0.05$, * $P \leq 0.05$, ** $P \leq 0.01$.** **DF)** Success Rate (SR %) of the proposed strategy for the different values of stiffness K.

P. 10, Line 481:

Statistical Analysis

The statistical analysis reported in the Results chapter quantifies the performance and the repeatability of our solution. To obtain the distributions of data that we used for the statistical analysis of the performance for varying stiffnesses (see Results section), from each experiment, we considered, we only consider the interval of time when interaction occurs (first contact to detachment) and extracted a single value of absolute tracking error of lateral position, an average value of longitudinal velocity, and a max value of interaction force, for each experiment. We define this interval as $[1, M]$ with M equal to the number of samples during the interaction, and extract the following metrics:

- The tracking error of lateral position over the interaction interval, by computing the mean absolute error (MAE): $e_p = \frac{1}{M} \sum_{k=1}^M |p_{y,k} - p_{y,ref}|$. This metric describes how close the drone is to the path.
- The tracking error of longitudinal velocity during the interaction interval, using the MAE as above: $e_v = \frac{1}{M} \sum_{k=1}^M |v_{x,k} - v_{x,ref}|$. This metric is expected to deviate from zero during the interaction, due to the response of the environment impeding the drone's motion.
- The interaction force along the longitudinal axis, by saving the maximum value during the interaction interval: $\max(|F_{ext,x,k}|)$ with $k = [1, M]$. This metric is connected to stability as it tracks high forces that may lead to undesired behaviors.
- The amplitude of the oscillations occurring on the attitude of the drone; we define this quantity by computing the RMS (root mean square) for both roll ϕ and pitch θ : $RMS(\phi) = \sqrt{(\sum_{k=1}^M (\phi_k)^2)/M}$, $RMS(\theta) = \sqrt{(\sum_{k=1}^M (\theta_k)^2)/M}$. This metric also refers to stability as it shows whether the drone is subject to abrupt oscillations on the attitude during the interaction.

Thus, ~~upon~~ for N experiments, we ~~had~~ have a distribution composed of N values for the introduced metrics. The Mann-Whitney U test was performed in MATLAB R2021a (MathWorks, MA, USA).

- The ablation study clearly demonstrates the importance of both the morphology and haptic feedback control. However, it is unclear why the controller alone sometimes succeeds for low stiffness but always fails for higher stiffness. Some discussion on the limitations of the controller by itself would strengthen this ablation study.

Response:

The ablation study examines a scenario in which the body is streamlined and has low friction, but controller is used without haptic feedback (Fig. 6C). With haptic feedback off, the controller enables the drone to follow the reference straight trajectory, but the impedance term in the controller is not active due to the absence of haptic feedback. The role of such a term is to appropriately suppress oscillations caused by the elastic response of the obstacle without the need for a model. For low stiffness obstacles, trajectory tracking, coupled with the low friction surface and streamlined shape of the shell, is sufficient for successful traversal. However, as the stiffness of the obstacle increases, the lack of haptic-driven control causes the drone to fail. In this condition the “haptically blinded” drone can neither use the feedback to push to traverse the obstacle nor dampen the oscillations caused by stiff obstacles and ultimately loses stability.

We have clarified this point in the “Ablation Study” section and made the drawings clearer at the top of Fig. 6.

Revision:

P. 6, Line 286:

~~Last, we conducted~~ Lastly, we conduct experiments to demonstrate the importance of ~~sensing the external wrench~~ haptic sensing to inform the feedback control loop, ~~for both monitoring~~ both to monitor the effects of the wrench on the ~~dynamics of the drone~~ drone’s dynamics and for reacting to the interaction with the environment during traversal. ~~In detail, we performed experiments neglecting the external wrench~~ To this end, we disable the haptic sensing, thereby removing the wrench measurements from the dynamics model and ~~removing the desired~~ the impedance term in the NMPC formulation, ~~i.e. treating the interaction~~. This allows the controller to track the reference trajectory, but the interaction is treated only as a disturbance (test for mid stiffness in 6C). For *low* stiffness ~~the task was accomplished successfully~~ obstacles, trajectory tracking, coupled with the low friction surface and streamlined shape of the shell, is sufficient for successful traversal due to the ~~light~~ weak mechanical response of the environment, ~~yet for~~ (small wrenches involved and negligible oscillations). However, as the stiffness of the obstacle increases, the lack of haptic-driven control causes the drone to fail. For *mid* and *high* ~~the drone failed~~ stiffness, the drone fails to traverse the obstacle (reaching instability) in 70% and 100% of the experiments respectively (success rate in Fig. 6D), due to the higher wrenches and notable oscillations. These experiments ~~show how the robot cannot rely only~~ demonstrate that the drone cannot rely solely on morphology-enabled behaviours to successfully traverse stiffer obstacles, ~~and confirm the~~. They confirm the crucial role of the ~~controller in opportunistically dampen~~ closed-loop haptic feedback controller in sensing and dampening the oscillations induced by the ~~obstacle~~ compliant obstacles during the interaction.

P. 20:

• The real branch experiments provide a nice proof-of-concept, but lack the detailed analysis done for the hinged plate experiments. It would be good to provide more quantification of the modes of interaction and forces/torques for the branch experiments too.

Response:

Thank you for your comment. We have extended the section “Single Branch Traversal” by providing quantitative results for the experiments with real branches. We used the same metrics introduced and evaluated for the experiments with the hinged surface to quantify the performance of the system in terms of path following and stability. The results confirm that it is difficult to clearly distinguish modes in these cases due to the short interaction time and the small forces involved.

Revision:

P. 6, Line 305:

~~Last, we performed experiments with~~Experiments are conducted using real branches (Fig. 7 and Supplementary Movie S4). The drone successfully ~~traversed~~traverses two branches - one branch without leaves (Fig. 7A) and the other one with small twigs and leaves (Fig. 7B) – both attached to a fixed structure ~~in~~at a single anchor point. ~~By considering the second branch as an example (Fig. 7C)~~ The performance of the system is quantified using the evaluation metrics introduced in the previous analysis (reported in the Methods chapter). Table 1 highlights that errors in lateral position remain close to zero across experiments. The errors in longitudinal velocity, the longitudinal force, and the oscillations on the pitch angle, are smaller compared to the experiments with the foam plate. This

reduction is attributed to the short interaction time and the rapid snapping away of the branches. The elongated shape and constraints of the branches results in a shorter and weaker interaction, ~~a clear distinction between multiple modes is harder to find due to the unstructured nature of the obstacle and~~ leading to a quick Push-and-Slide mode. This is evident in Fig. 7C, illustrating ~~the shorter interaction time, yet it is possible to see the~~ usage of Push-and-Slide mode due to ~~the increase of both~~ increased longitudinal and lateral forces $F_{ext,x}$ and $F_{ext,y}$, and ~~the~~ torque around the vertical axis $\tau_{ext,z}$.

P 23:

Metrics	Test 1	Test 2	Test 3	Test 4
e_p [m]	0.066	0.063	0.044	0.033
e_p [m/s]	0.104	0.044	0.078	0.069
$\max(F_{ext,x})$ [N]	0.250	0.211	0.182	0.239
$RMS(\phi)$ [deg]	1.615	1.325	1.299	0.742
$RMS(\theta)$ [deg]	2.875	3.471	1.134	1.695

Table 1: **Quantitative results of the experiments with real branches.**

REVIEWER COMMENTS

Reviewer #1 (Remarks to the Author):

The authors have well addressed my concerns. This manuscript is acceptable.

Reviewer #2 (Remarks to the Author):

I want to thank the authors for carefully revising the paper based on my comments. I have a few remaining questions about the responses the authors submitted.

1. The authors used a custom-designed simulator to run ablation studies and fine tune the NMPC. However, there is not much discussion or verification about the simulator, other than stating it was built using Gazebo and ROS. Since the new ablation studies were completely based on this simulator, the authors should comments on the possible gaps in sim-to-real transfer. Also has the team open-sourced this simulator or plan to do so?

2. In the ablation study results (Supp. Figure 3), the authors reported the statistics among five trials. I am wondering where was the uncertainty/variance introduced into the simulation studies across the five trials in each condition?

Response to Reviewers

Synergistic Morphology and Feedback Control for Traversal of Unknown Compliant Obstacles with Aerial Robots

Emanuele Aucone, Christian Geckeler, Daniele Morra, Lucia Pallottino, and Stefano Mintchev

Dear Reviewers,

We would like to express our gratitude for the valuable review process. The manuscript has been revised according to the suggested reviews.

Best regards,

Emanuele Aucone, Christian Geckeler, Daniele Morra, Lucia Pallottino, and Stefano Mintchev

Reviewer #2

I want to thank the authors for carefully revising the paper based on my comments. I have a few remaining questions about the responses the authors submitted.

1. The authors used a custom-designed simulator to run ablation studies and fine tune the NMPC. However, there is not much discussion or verification about the simulator, other than stating it was built using Gazebo and ROS. Since the new ablation studies were completely based on this simulator, the authors should comments on the possible gaps in sim-to-real transfer. Also has the team open-sourced this simulator or plan to do so?

Response:

We thank the reviewer for the question. As follows, we explain the process of setting up our simulation environment, we perform a comparison of the performances in simulation and real-world experiments, and we discuss about gaps and consequent mitigations.

We designed our drone developing a CAD model and specified the material properties of each component. Then we built the drone, we measured the real mass of the drone, and we noticed a difference in terms of weight. This is to be expected due to mechanical inaccuracies associated with the manufacturing process, as well as additional cabling and assembly of the components. Thus, we estimated the moments of inertia of the real drone by linearly scaling the inertias output from the CAD, using the ratio between the real mass and the one from the CAD as scaling factor. This is valid as we assume that the increase in the mass is uniformly distributed over all the components.

Therefore, we set up the simulation environment using Gazebo, a 3D physics-based simulator, and RotorS package, which allows to simulate drone models and their dynamics. We included a Force/Torque (F/T) sensor with a plugin that further simulates measurements with a Gaussian noise, as well as the 3D model of the robot's cage, exported from the CAD. For the state estimation of the drone, we used the ground truth odometry from RotorS and Gazebo. Hence, we defined mass and inertia of each body (URDF of the quadrotor, the F/T sensor, and the cage), by using the real masses and the scaled inertias mentioned above. This setup, which runs on ROS, was useful to develop the controller pipeline, which was then transferred to the drone. The real robot uses a load cell as F/T sensor and a tracking camera for Visual-Inertial Odometry (VIO).

The main gaps between simulation and reality relate to model uncertainties (in terms of parameters mismatch due to the estimated inertia), sensors readings inaccuracy (as the load cell has a different type of noise and the tracking camera is less accurate that the ground truth of the sim), and eventually additional disturbances (due to aerodynamics effects experienced during flight). To assess these gaps, we performed a quantitative analysis by comparing the performances in simulated and real-world experiments (Supplementary Fig. 5 and Supplementary Tab. 3). For evaluation, we used the performance metrics defined in the Methods chapter and the experiments where the set of balanced weights was used for both sim and real. To have the same number of samples in the distribution of data we extended the simulation tests. We revised the article by including these results, which confirm that there are sim-to-real gaps yet they are relatively small, and we further present a discussion to properly explain the gaps and how to mitigate them.

To conclude, we can open source the simulator, we will add it to the supplementary material attached with the article.

Revisions:

P. 23, Line 589:

... has a total mass of 1.12 kg. This value has been measured with an accurate scale, and it differs from the one computed by the CAD, which can happen due to mechanical inaccuracies associated with the

manufacturing process, as well as additional modelling inaccuracies (e.g. cabling). The moments of inertia of the real system have been estimated by linearly scaling the inertias output from the CAD, using the ratio between the real mass and the one from the CAD as scaling factor. This assumes that the extra mass is uniformly distributed on all the components, without a change in the dimension, therefore linearly affecting the moments of inertia. It is worth noticing that high model inaccuracies may generally degrade the performance of model-based controllers. Inertial parameters useful for the dynamical model inside the NMPC are ~~measured (mass) or obtained from the CAD (moments of inertia)~~, and reported in Supplementary Tab. 1.

P. 35:

Inertial Parameters	Values
Mass [kg]	1.1004 .200
Inertia along x-axis [kg m ²]	0.007
Inertia along y-axis [kg m ²]	0.007
Inertia along z-axis [kg m ²]	0.012

Supplementary Table 1: Inertial parameters of the designed aerial robot.

P. 23, Line 617:

We developed and validated the NMPC and the overall strategy using the Robot Operating System (ROS) and Gazebo 3D simulator (Supplementary Fig. 2A). We used the RotorS [1] package for simulating a quadrotor, ~~as well as its dynamics and its-flight architecture, to which we added an; then, we incorporated a F/T sensor with a plugin that simulates measures with Gaussian noise; finally, we included the 3D model and the CAD~~ of the cage ~~exported from the CAD~~ – the visual model of the cage is a sphere but the interaction occurs only on the ring and the inertial parameters corresponds to ~~the ones used for~~ the real-world prototype (Supplementary Tab. 1). We used a Gazebo plugin to generate an elastic behavior of the environment, by using one rotational spring joint. We performed several experiments in the 3D physical simulator for implementation, debug, tuning and evaluation. ~~A study on the sim-to-real gaps is reported in a dedicated section (see Sec. Sim-to-Real Gaps).~~

P. 24, Line 638:

At runtime, for each iteration, the NMPC receives the data from the sensors (i.e. the external wrench and the state), sets the initial state condition using the current values of the state, and integrates the continuous dynamics in order to discretize it. ~~Eventual inaccuracies in the sensor readings may directly affect the evolution of the dynamics, so it is important to be sure that the sensors used onboard are accurate enough to allow to find a feasible solution of the NMPC.~~ In our approach, the only physical parameters required for the dynamics of the system used in the optimization are the inertial properties of the drone (mass and moments of inertia, Supplementary Table 1). ~~Generally, inaccuracy in the mass just results in a difference of the thrust force that the drone has to generate, whereas mismatches in the inertia, thus on the rotational dynamics, may result in degraded performances of the controller.~~ Since the drone operates in quasi-static motion and the main contribution in the rotational dynamics (last row of Eq. 1) comes from the external torques, variations on the moments of inertia have a small impact during the interaction. ~~We experimentally validate this and report the results in Sec. Sim-to-Real Gaps. Accordingly, inaccuracy in the mass just results in a difference of the thrust force that the drone has to generate.~~

P. 26, Line 789:

Sim-to-Real Gaps - Analysis and Discussion

To quantitatively assess the gaps between our custom simulation environment and the real-world setup, we performed an analysis that compares the performances between simulated and real-world experiments in terms of the 5 metrics defined in the Methods chapter. For evaluation, the experiments

with the set of balanced weights were used for both sim and real. To have the same number of samples in the distribution of data we extended the simulation tests. The statistical analysis is reported in Supplementary Fig. 5, whereas Supplementary Table 3 summarizes the study by considering the average value within the distribution of data, for each metric, to have a more comprehensive view of our findings.

In terms of interaction, the gap in the longitudinal forces is small (Supplementary Fig. 5C), with the highest discrepancy for low stiffness (Supplementary Tab. 3). This was expected as we simulated a F/T sensor with Gaussian noise, yet the load cell mounted on the real drone might have different types of additive noise and non-static biases that drift over time. Another reason that creates a gap relates to the fact that on the real drone the sensor is affected by strong vibrations due to the attachment with the rigid frame of the drone, as well as by aerodynamic effects of the airflow on the cage, which is placed right below the propellers. Such effect, which is missing in simulation, introduces additional disturbance on the sensor readings. It is worth mentioning that in our study contact force models are not included in the NMPC prediction model, since the interaction is with an unknown environment (as discussed in the Suppl. Info.). The interaction, indeed, relies on the impedance term that directly exploits the measure of the sensor. This means that both in sim and real-world experience, despite the noise might be different, the impedance term allows the NMPC to operate under uncertainties by using a reactive behavior (as discussed in the Suppl. Info. and in [9]).

In terms of tracking performance, instead, the effect of having state estimator between simulation and reality is more evident in the results, despite small gaps. The real drone, indeed, uses a tracking camera running VIO onboard – the signals from this sensor are less accurate than the simulated ground truth. Generally, the consequence of having a tracking camera (which can be affected by noise as well as drifts, especially in variables like velocity which is derived from position) is quite more evident on predictive controllers w.r.t. reactive ones (as in the previous case), given the fact that a wrong state estimation generates an erroneous evolution of the model internally simulated and, in turn, a misleading control input that finally produces an inaccurate trajectory tracking [10]. Supplementary Tab. 3 shows similar errors in terms of tracking the lateral position, which is expected as the drone needs to slightly drift from the reference path while sliding on it (more evident for increasing stiffness). In terms of variability of these errors, Supplementary Fig. 5A confirms that the ground truth of the simulation ensures a more repeatable behavior, against the high dispersion of data for real-world experiment, which is due to the inaccuracies of the state estimation itself. Tracking performance in terms of velocity errors are more visible in terms of average values (Supplementary Tab. 3) as well as statistical difference (Supplementary Fig. 5B, P value < 0.01). The gap, however, is in the order of 0.01 m/s. Tracking performance can be enhanced by using a motion-capture system that provides estimation of position, velocity, and orientation of the drone with an accuracy below the millimeter.

The oscillations on the attitude (both roll and pitch) exhibit the highest difference between sim and real-world. The statistical analysis highlights a difference with a significance of 99% (Supplementary Fig. 5D, E), as the oscillations on the attitude are limited to values below 1 degree. Despite this gap, Supplementary Tab. 3 reminds that the oscillations experienced on the real drone are limited to a few degrees, lower than the safety limits imposed by the NMPC ([-20,20] deg). A less accurate state estimate for the real-world scenario leads to measurements that present some noise and a higher dispersion also in this case (Supplementary Fig. 5D, E), as the attitude is provided by the VIO. Further, we believe that the higher values of oscillations on the real drone also relate to inaccuracies on the inertia between simulated and real drone, which directly affects the rotational dynamics. Sensor-based controllers use instantaneous sensor measurement, instead of an explicit model, to represent system dynamics, thus they are more robust against model uncertainties and external disturbances. It is known instead that model uncertainties in the form of unmodeled complex aerodynamic effects, varying payloads and parameter mismatch can degrade the overall system performance for model-based controllers.

To conclude, the analysis on the sim-to-real gap has helped to identify gaps, which highlight that the simulation is more conservative and it better ensures the quasi-static motion, thanks to a higher accuracy of the sensors used on-board, as well as a perfect modeling of the dynamics' parameters used by the NMPC. Our objective was not to have a high-fidelity simulator for perfect sim-to-real transfer, but rather to exploit it to validate the idea of the combination of body morphology and minimalistic controller for the task of traversing compliant obstacles. In our case, the gaps have been shown to be small enough to avoid a severe degradation of the performance of the system after the transfer from simulation. In the future, a first direction relates to estimating the real inertia of the drone, which can surely help to make the model-based controller more robust and achieve better performance. Further, to compensate for additional uncertainties and disturbances on the rotational dynamics, the uncertainties could be considered as a variable in the rotational dynamics' equations as a safety measure [11], they could be further estimated online to make the MPC adaptive [12], or a downstream controller could be added to regulate the original NMPC command [13]. These approaches have been validated for free-flight at high speed (not for interaction tasks), yet have proved to improve both robustness to mismatches and tracking performance.

P. 34:

Supplementary Figure 5: **Assessment of the sim-to-real gaps for experiments conducted with balanced weight.** **A** Tracking error of the lateral position; **B** longitudinal velocity error; **C** max longitudinal force; **D** oscillations on the roll motion; **E** oscillations on the pitch motion. Mann-Whitney U test performed among the two sets for all the stiffnesses; box plots indicate median (middle line), 25th, 75th percentile (box) and 5th and 95th percentile (whiskers) as well as outliers (single points);

number of tests used for the statistical analysis = 9 for each stiffness; (no asterisk) $P > 0.05$, * $P < 0.05$, ** $P < 0.01$.

P. 37:

	Low stiffness		Mid stiffness		High stiffness	
	Sim	Real	Sim	Real	Sim	Real
Pos err [m]	0.051	0.0945	0.104	0.110	0.141	0.088
Vel err [m/s]	0.035	0.108	0.040	0.101	0.045	0.110
Max force [N]	0.700	1.292	1.154	1.394	1.592	1.719
Roll oscill [deg]	0.174	2.973	0.303	2.497	0.443	3.010
Pitch oscill [deg]	0.405	4.029	0.498	5.138	0.611	5.833

Supplementary Table 3: Average value for the 5 metrics used to compare the performances in simulation and real-world experiments.

Additional References:

- [9] A. Alharbat, H. Esmaeeli, D. Bicego, A. Mersha and A. Franchi, "Three Fundamental Paradigms for Aerial Physical Interaction Using Nonlinear Model Predictive Control," 2022 International Conference on Unmanned Aircraft Systems (ICUAS), Dubrovnik, Croatia, 2022, pp. 39-48, doi: 10.1109/ICUAS54217.2022.9836221.
- [10] Bicego, D., Mazzetto, J., Carli, R. et al. Nonlinear Model Predictive Control with Enhanced Actuator Model for Multi-Rotor Aerial Vehicles with Generic Designs. *J Intell Robot Syst* 100, 1213–1247 (2020). <https://doi.org/10.1007/s10846-020-01250-9>.
- [11] S. Sun, A. Romero, P. Foehn, E. Kaufmann and D. Scaramuzza, "A Comparative Study of Nonlinear MPC and Differential-Flatness-Based Control for Quadrotor Agile Flight," in *IEEE Transactions on Robotics*, vol. 38, no. 6, pp. 3357-3373, Dec. 2022, doi: 10.1109/TRO.2022.3177279.
- [12] D. Hanover, P. Foehn, S. Sun, E. Kaufmann and D. Scaramuzza, "Performance, Precision, and Payloads: Adaptive Nonlinear MPC for Quadrotors," in *IEEE Robotics and Automation Letters*, vol. 7, no. 2, pp. 690-697, April 2022, doi: 10.1109/LRA.2021.3131690.
- [13] F. Nan, S. Sun, P. Foehn and D. Scaramuzza, "Nonlinear MPC for Quadrotor Fault-Tolerant Control," in *IEEE Robotics and Automation Letters*, vol. 7, no. 2, pp. 5047-5054, April 2022, doi: 10.1109/LRA.2022.3154033.

2. In the ablation study results (Supp. Figure 3), the authors reported the statistics among five trials. I am wondering where was the uncertainty/variance introduced into the simulation studies across the five trials in each condition?

Response:

As mentioned in the previous answer, we simulate the F/T sensor with a Gaussian noise, we don't use the real forces between bodies from the physics engine. This introduces variability between the tests, as the response of the controller (thus, of the system) will be different during each experiment.

Revisions:

P. 23, Line 617:

We developed and validated the NMPC and the overall strategy using the Robot Operating System (ROS) and Gazebo 3D simulator (Supplementary Fig. 2A). We used the RotorS [1] package for

simulating a quadrotor, as well as its dynamics and its flight architecture, to which we added an; then, we incorporated a F/T sensor with a plugin that simulates measures with Gaussian noise; finally, we included the 3D model and the CAD of the cage exported from the CAD – the visual model of the cage is a sphere but the interaction occurs only on the ring and the inertial parameters corresponds to the ones used for the real-world prototype (Supplementary Tab. 1). We used a Gazebo plugin to generate an elastic behavior of the environment, by using one rotational spring joint. We performed several experiments in the 3D physical simulator for implementation, debug, tuning and evaluation. A study on the sim-to-real gaps is reported in a dedicated section (see Sec. Sim-to-Real Gaps).

P. 25, Line 721:

To quantitatively assess the impact of the controller parameters on the performance of the system in terms of path following and safe physical interaction – the two main objectives of the NMPC – we perform an ablation study in simulation for different weight parameters included in the optimization cost707 vectors (Supplementary Fig. 3 and Supplementary Movie S5). Variability among the tests is ensured using simulated noise on the F/T sensor, as the controller will receive different sensor readings during each experiment and consequently have a different behavior depending on the interaction forces.

Extra:

Regarding the statistics for the ablation study, we apologize for an error that we found in one script about the simulation. There was a negative sign instead of a positive one for the reference velocity, and an absolute value missing for the force, before applying the max. We updated Fig. S3B, the errors in velocity are smaller than before, but the discussions about these specific results are still valid, both in terms of statistical analysis and in terms of the effect of the NMPC weights on the velocity error. Same applies for the discussion on the forces.

REVIEWERS' COMMENTS

Reviewer #2 (Remarks to the Author):

I'd like to thank the authors for carefully revising the paper based on my last round of comments. I do not have additional comments.